# Proliferation-independent regulation of organ size by Fgf/Notch signaling

**Agnė Kozlovskaja-Gumbrienė[1], Ren Yi[1†], Richard Alexander[1], Andy Aman[1‡], Ryan Jiskra[1], Danielle Nagelberg[2], Holger Knaut[2], Melainia McClain[1], Tatjana Piotrowski[1]\***

[1]Stowers Institute for Medical Research, Kansas City, United States; [2]Developmental Genetics Program and Kimmel Center for Stem Cell Biology, Skirball Institute of Biomolecular Medicine, New York University Langone Medical Center, New York, United States

**Abstract** Organ morphogenesis depends on the precise orchestration of cell migration, cell shape changes and cell adhesion. We demonstrate that Notch signaling is an integral part of the Wnt and Fgf signaling feedback loop coordinating cell migration and the self-organization of rosette-shaped sensory organs in the zebrafish lateral line system. We show that Notch signaling acts downstream of Fgf signaling to not only inhibit hair cell differentiation but also to induce and maintain stable epithelial rosettes. Ectopic Notch expression causes a significant increase in organ size independently of proliferation and the Hippo pathway. Transplantation and RNASeq analyses revealed that Notch signaling induces apical junctional complex genes that regulate cell adhesion and apical constriction. Our analysis also demonstrates that in the absence of patterning cues normally provided by a Wnt/Fgf signaling system, rosettes still self-organize in the presence of Notch signaling.

\*For correspondence: pio@stowers.org

**Present address:** [†]Center for Genomics and Systems Biology, Department of Biology, New York University, New York, United States; [‡]Department of Biology, University of Washington, Seattle, United States

**Competing interests:** The authors declare that no competing interests exist.

## Introduction

Organ morphogenesis relies on the integration of complex processes, such as cell migration, cell shape changes and cell specification to generate the correct three-dimensional geometry necessary for function. Additionally, these cell behaviors must be coupled with mechanisms that regulate the final size of the organ for correct integration into the organism. Understanding the mechanisms that underlie these phenomena is a major focus of modern biology. Even though the intracellular mechanisms leading to cell shape changes are fairly well understood, we are only beginning to elucidate how signaling pathways coordinate cell shape changes with the development of a whole tissue or embryo. Additionally, the importance of multicellular, epithelial rosettes in *Drosophila* axis elongation and retina development, mouse pre-implantation embryo morphogenesis, pancreas development, brain tumors or the neural stem cell niche has only fairly recently been recognized (*Bedzhov and Zernicka-Goetz, 2014*; *Blankenship et al., 2006*; *Harding et al., 2014*; *Martin and Goldstein, 2014*; *Wippold and Perry, 2006*).

The zebrafish lateral line is a powerful model to study sensory organ morphogenesis, as it develops superficially in the skin and is amenable to experimental manipulation and in vivo imaging. The lateral line is a sensory system for the detection of water movements and consists of rosette-shaped sensory organs (neuromasts) that are arranged in lines along the body of the animal. Each neuromast is composed of sensory hair cells surrounded by support cells. Lateral line hair cells are homologous to vertebrate inner ear hair cells and are specified by the same molecules (*Nicolson, 2005*). The lateral line system on the trunk develops from an ectodermal placode posterior to the ear that migrates to the tail tip. The migrating placode (now called primordium) periodically deposits clusters

of cells that mature into neuromasts, thus forming a line of sensory organs. This migrating primordium consists of a mesenchymal leading region and a trailing region in which cells apically-basally polarize and then apically constrict to form garlic bulb/rosette-shaped proneuromasts (reviewed in [*Harding et al., 2014*]). The two domains are maintained by a feedback mechanism between the Wnt and Fgf pathways (*Aman and Piotrowski, 2008*). Activation of the Wnt pathway in the leading region induces Fgf ligands that activate the Fgf pathway in the trailing region. Fgf ligand expression is uniform in the leading region but then becomes restricted to one central cell as organs (proneuromasts) begin to form. Fgf ligand expression by a central proneuromast cell that activates Fgf signaling in surrounding cells is crucial for proneuromast formation and maintenance (*Durdu et al., 2014*; *Ernst et al., 2012*; *Harding and Nechiporuk, 2012*; *Lecaudey et al., 2008*; *Nechiporuk and Raible, 2008*). Fgfr-Ras-Mapk signaling is thought to directly induce apical constriction and rosette formation via the activation of the actin-binding protein *shroom3*. *shroom3* leads to apical localization of Rock2a kinase that phosphorylates non-muscle myosin II (pNMII) driving actomyosin constriction (*Ernst et al., 2012*; *Harding and Nechiporuk, 2012*).

Of all the mutants/manipulations thus far analyzed that affect lateral line development only very few lead to an increase in organ size. The only manipulation described that causes an increase in neuromast size is the upregulation of Wnt signaling, while the size of the primordium increases after inhibition of the Hippo pathway member *amotl2a* (*Agarwala et al., 2015*; *Head et al., 2013*; *Wada et al., 2013*; *Wada and Kawakami, 2015*; *Jacques et al., 2014*). The Hippo pathway controls organ size via a kinase cascade that leads to the phosphorylation and degradation of the transcriptional co-activators Yap/Taz (*Sun and Irvine, 2016*). In the absence of pathway activation, Yap/Taz are translocated to the nucleus where they activate proliferation and survival genes. The Wnt pathway affects organ size via controlling proliferation, which, at least partially, is regulated by *amotl2a* (*Agarwala et al., 2015*). In contrast, loss of the transcriptional co-activator *yap1* rescues the *amotl2a* overproliferation phenotype but does not interact with the Wnt pathway. Thus, the details of how Wnt, *amotl2a* and *yap1* are integrated to affect proliferation are not well understood. Here we describe how the upregulation of Notch signaling by overexpression of the Notch1a intracellular domain (ICD; NICD, [*Scheer and Campos-Ortega, 1999*]) in the primordium, leads to a significant increase in neuromast size. The increased neuromast size in both Notch and Wnt overexpressing embryos is *yap1*-independent, and in case of NICD the organ size increase is even proliferation-independent.

Apical constriction is driven by cortical tension-generating actomyosin contractions that are transmitted to neighboring cells via apically localized cadherins (reviewed in [*Heisenberg and Bellaïche, 2013*; *Martin et al., 2009*; *Pilot and Lecuit, 2005*; *Schwayer et al., 2016*]). We demonstrate that Fgf-dependent Notch signaling cell-autonomously, and downstream of Fgf signaling upregulates adherens and tight junction molecules. These molecules together form an apical junctional complex associated with a circumferential actomyosin belt (*Niessen, 2007*). Thus, Notch signaling increases organ size independently of proliferation by coordinating actomyosin activation and cell adhesion.

## Results

### Constitutive activation of Notch or Wnt signaling generates larger sensory organs

When we constitutively activate Notch signaling in the lateral line by driving the NICD in (*Tg(cldnB:lynGFP)*;*Tg(cldnB:gal4) x Tg(UAS:nicd)*) embryos, we observe that deposited neuromasts are substantially larger than in sibling embryos (*Figure 1A–B'*, *Figure 1—figure supplement 1A–A'*; *Video 1*). The NICD transgene is strongly induced when the lateral line placode is forming at 13 hr post fertilization (hpf) (data not shown). It is also myc-tagged and and shows widespread expression in the lateral line and skin (*Figure 1C–D*). The neuromast size increase is reminiscent of Wnt overexpressing *apc* mutant neuromasts (*Figure 1F–F'*, [*Wada et al., 2013*]). In contrast, a reduction of Notch signaling in *mib1* mutants causes fragmentation of neuromasts (*Figure 1E,E'*; *Video 2*; [*Matsuda and Chitnis, 2010*]).

Labeling of the first deposited trunk neuromast (L1) with the actin marker phalloidin shows that the NICD apical F-actin meshwork is larger compared to wildtype neuromasts, suggesting that NICD neuromasts are composed of more apically constricted cells (*Figure 1G–H'*). The robust induction of

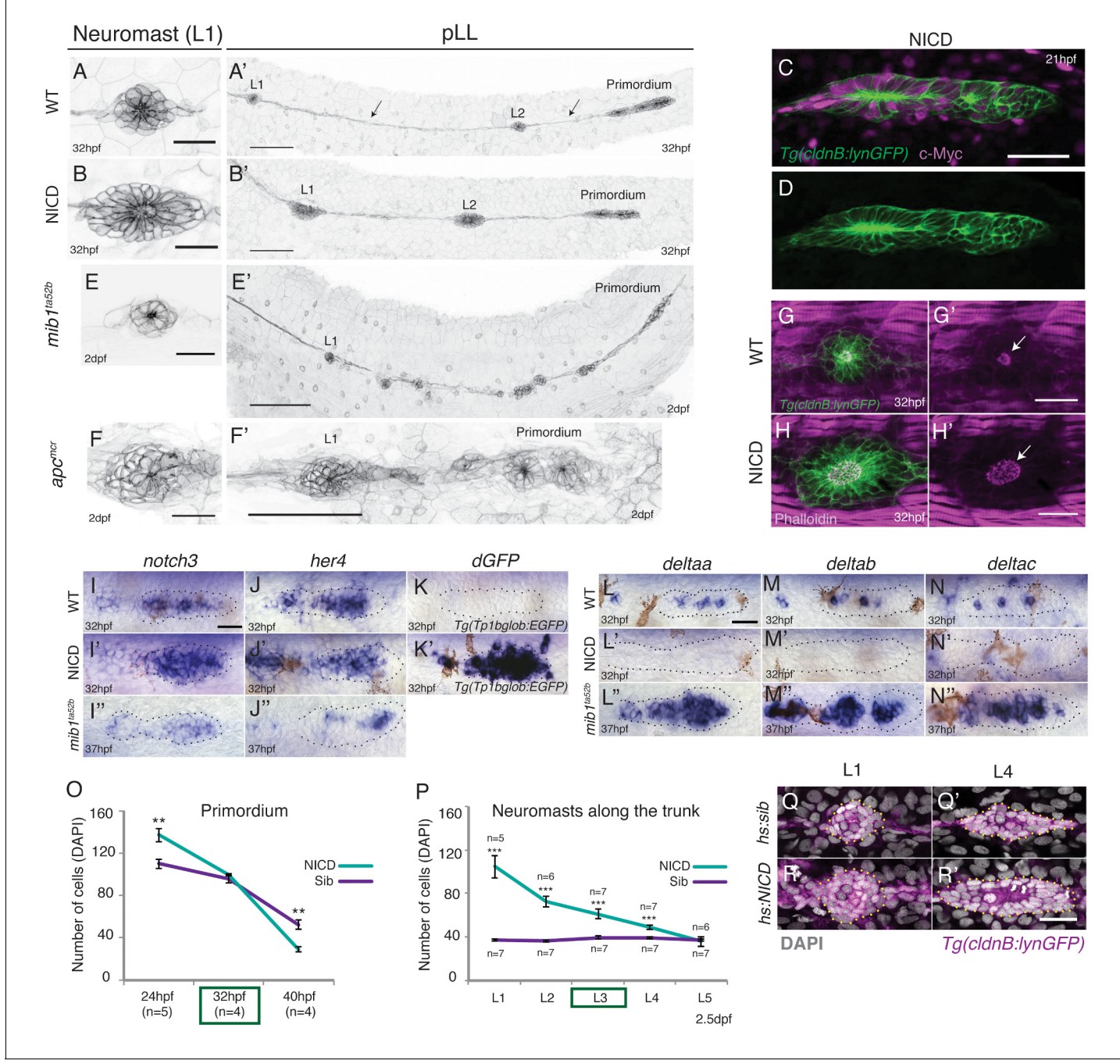

**Figure 1.** Induction of Notch signaling induces larger lateral line organs. (**A** and **B** and **E** and **F**) First deposited trunk neuromasts (L1) in *Tg(cldnB: lynGFP)*. (**B**) NICD and (**F**) *apc^{mcr}* neuromasts are larger than (**A**) WT and (**E**) *mib1^{ta52b}* neuromasts. Scale bar is 25 µm. (**A'** and **B'** and **E'** and **F'**) Posterior lateral line in *Tg(cldnB:lynGFP)*. (**A'**) WT, (**B'**) NICD, (**E'**) *mib1^{ta52b}* and (**F'**) *apc^{mcr}*. Scale bar is 100 µm. (**C** and **D**) NICD transgenic embryos are labeled by a Myc-tag. The c-Myc antibody in magenta. Scale bar is 50 µm. (**G–H'**) L1 neuromasts stained with Phalloidin at 32 hpf. In NICD neuromasts (**H** and **H'**) the apical F-actin meshwork (arrow) is larger compared to wildtype (WT) (**G** and **G'**), suggesting that NICD neuromasts are composed of more apically constricted cells. Scale bar is 25 µm. (**I–N''**) Expression of Notch pathway genes in the primordium. (**I'**) *notch3*, (**J'**) *her4* and *dgfp* in the Notch reporter *Tg(Tp1bglob:EGFP)* (**K'**) are upregulated in NICD primordia compared to WT (**I** and **J** and **K**) and *mib1^{ta52b}* (**I''** and **J''**) primordia. The Notch reporter only seems to be activated by high levels of Notch signaling, as it is not active in wildtype primordia (**K**), even though the Notch target *her4* is expressed (**J**). Scale bar is 25 µm. (**L–N''**) Expression of *delta* ligands in the primordium. (**L'**) *deltaa*, (**M'**) *deltab* and *deltac* (**N'**) are largely downregulated in NICD compared to WT (**L** and **M** and **N**) and *mib1^{ta52b}* (**L''** and **M''** and **N''**) primordia. Scale bar is 25 µm. (**O**) NICD and sibling primordia sizes decrease overtime. NICD primordia start out (24 hpf) with significantly more cells compared to sibling primordia, but after the second deposition cycle at ~32 hpf NICD and sibling primordia have a similar amount of cells. By 40 hpf NICD primordia are composed of significantly less

*Figure 1 continued on next page*

*Figure 1 continued*

cells. Error bars represent standard error (p<0.01=** Student's *t* test). (**P**) NICD neuromasts (L1-4) along the trunk of the embryo at 2.5 dpf consist of significantly more cells compared to siblings. NICD primordia deposit big neuromasts, even at 32 hpf when NICD and sibling primordia possess the same cell number (time point (32 hpf) is marked in the (**O**) green box, also see L3 deposition in the *Video 3*). Error bars indicate standard error (p<0.001=*** Student's *t* test). (**Q–R'**) Notch activation in *hs:NICD* embryos after the primordium and ganglion have separated still significantly increases neuromast sizes (**R'**) compared to a (**Q'**) sibling (see *Figure 1—figure supplement 1E* for quantification). These results indicate that the number of cells in neuromasts is independent of primordium size. Notch overexpression was induced by a 39°C heat-shock for 45 min starting at 25 hpf (L1 was still a part of the primordium). Embryos were fixed after L4, L5 deposition (~40 hpf). Scale bar is 25 μm.

The following figure supplement is available for figure 1:

**Figure supplement 1.** NICD primordia do not migrate to the tail tip and deposit fewer neuromasts.

NICD in (*Tg(cldnB:lynGFP);Tg(cldnB:gal4) x Tg(UAS:nicd)*) transgenic embryos, is also evidenced by upregulation of the Notch targets *notch3*, *her4*, strong activation of the Notch reporter *Tp1bglob: EGFP* and downregulation of *delta* ligands (*Figure 1I–K'* and *Figure 1L–N'*). On the other hand, in *mib1* mutants *notch3* and *her4* are reduced but not completely absent and *delta* genes are upregulated (*Figure 1I''–J''* and *L''–N''*; *Matsuda and Chitnis, 2010*). NICD primordia migrate more slowly and deposit fewer neuromasts than wildtype primordia and 80% of the NICD primordia do not reach the tail tip, as they deposit too many cells and run out of cells (*Figure 1O* and *Figure 1—figure supplement 1A–C*).

During early posterior lateral line development, Notch determines the proportion of placodal cells that contribute to the lateral line ganglion posterior to the ear and how many cells will become part of the migrating primordium (*Mizoguchi et al., 2011*). Consequently, ectopic activation of Notch signaling generates an initially larger primordium at the expense of the ganglion, which could explain the increase in neuromast size (*Figure 1O* and *Figure 1—figure supplement 1D*). Indeed primordium size limits how large NICD neuromasts can grow. NICD primordia become increasingly smaller during migration and NICD neuromasts accordingly decrease in size the further posterior they are located along the trunk of the embryo (*Figure 1O and P*). However, by 32 hpf NICD and sibling primordia consist of the same number of cells but NICD primordia still deposit larger L3 neuromasts (*Figure 1O and P*; *Video 3*). Also, heat-shock activation of Notch signaling after the primordium and ganglion have separated significantly increases the neuromast cell number (*Figure 1Q–R'*, *Figure 1—figure supplement 1E*). These results indicate that, although the primordium size determines the maximum size of an enlarged neuromast, the increased NICD neuromast size is independent of the early role of Notch in allocating cells to the primordium.

## The increased organ size after Notch and Wnt activation is independent of *yap1*

The size of the migrating primordium is partially controlled by the Hippo pathway components *amotl2a* (a tight junction-associated scaffolding protein) and *yap1* (a transcriptional co-activator) (*Agarwala et al., 2015*). *amotl2a* inhibits proliferation, whereas *yap1* is required in the primordium to reach its normal size. To test if the Hippo pathway might modulate neuromast size downstream of Notch or Wnt signaling we tested if (a) Manipulating Wnt or Notch signaling affects the expression of Hippo pathway genes and (b) if the downregulation of *yap1* rescues neuromast size in NICD and *apc* embryos. *yap1* is expressed in the leading 2/3 of the wildtype primordium (*Figure 2A'*) and is upregulated in NICD (*Figure 2B'*). *yap1* is also increased in *apc* primordia correlating with the increased proliferation observed in these embryos (*Figure 2D'*; [*Aman et al., 2011*]). In contrast, *yap1* is downregulated in *mib1* and Wnt depleted *hs:dkk1b* primordia (*Figure 2C',E'*). However, *yap1* is not

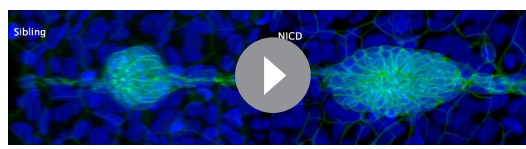

**Video 1.** 3D rendering of a 2 dpf sibling and NICD L1 neuromast using FluoRender image visualization software (University of Utah). *Tg(cldnB:lynGFP)* embryo was fixed in 4% PFA and stained with DAPI to visualize nuclei.

**Video 2.** Time-lapse movie of a *mib1^{ta52b}* mutant in the *Tg(cldnB:lynGFP)* background showing the gradual disintegration of neuromasts and the primordium. Embryo was imaged every 5 min starting at 28 hpf.

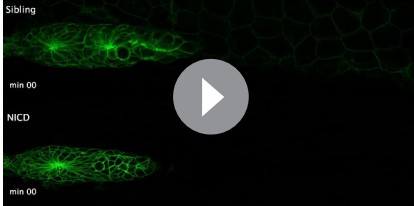

**Video 3.** Time-lapse movie showing the primordium depositing the L3 proneuromast in a sibling- *Tg(cldnB:lynGFP)* and a *Tg(cldnB:lynGFP);NICD* embryo. The NICD primordium deposits a larger proneuromast compared to the wildtype sibling. Embryos were imaged every 5 min.

expressed in deposited wildtype, NICD, *mib1*, *apc* or *hs:dkk1b* neuromasts suggesting that proliferation in deposited neuromasts is *yap1*-independent (*Figure 2A–E*). We also analyzed the canonical Hippo pathway inhibitor *stk3* (the ortholog of *hippo* kinase in *Drosophila*) and *amotl2a*, which inhibits Wnt-induced proliferation (*Figure 2—figure supplement 1A–J'*; *Agarwala et al., 2015*). The expression of these genes does not correlate with proliferation in the primordia with manipulated Wnt or Notch signaling, showing that we do not yet fully understand the integration of Hippo signaling with other mitogens and that further studies are needed.

However, the upregulation of *yap1* in primordia correlates with the increased neuromast cell numbers in *apc* and NICD. In addition, *yap1* morpholino injections inhibit the ectopic proliferation in the primordium that is induced by loss of *amotl2a*, phenocopying the *yap1* mutant (*Agarwala et al., 2015*). We therefore injected *yap1* morpholino into NICD and *apc* embryos and counted the number of cells in the deposited neuromasts, as well as migrating primordium. *yap1* morpholino reduces the number of primordium cells in its expression domain in the leading 2/3 of wildtype, *apc* and NICD primordia (*Figure 2F*). As morpholino injections can cause cell death in the lateral line (*Aman et al., 2011*), we injected the *yap1* morpholino into homozygous *p53* mutant embryos (*Figure 2—figure supplement 1K*). The primordium cell number decreases from approximately 100 to 80, which is similar to what we find in *yap1* morpholino-injected wildtype embryos (*Figure 2H*), suggesting that the decrease in cell number is not caused by morpholino-induced *p53* activation (*Kok et al., 2015*). Although *yap1* downregulation significantly reduces the size of the primordium, it does not rescue neuromast size in *apc* or NICD embryos to a wildtype level (*Figure 2G*). The reduction in NICD L1 and L2 pro-neuromast size in *yap1* morphants (*Figure 2F and G*) can be attributed to the role of *yap1* in primordium size regulation and that the primordium size limits the number of cells available for allocation into forming neuromast (*Figure 1O and P*). The above experiments show that the increase in neuromast size in *apc* and NICD is *yap1* independent.

## Activated Notch leads to proliferation-independent neuromast growth

Activation of Wnt signaling produces larger neuromasts via hyperproliferation and we therefore tested if proliferation is upregulated in a *yap1*-independent fashion in NICD primordia and neuromasts (*Wada et al., 2013*). In contrast to *apc* neuromasts, which show a significant increase in proliferation, NICD neuromasts show a significant decrease (*Figure 3A–D*). Accordingly, *apc* neuromasts gain a significant number of cells after deposition between 32–56 hpf, whereas NICD neuromasts do not significantly grow in cell number (*Figure 3E*). The BrdU index and number of primordium cells in 31 hpf *apc* primordia is not significantly higher than in the siblings (*Figure 3F,F',H,I*), although 33 hpf *apc* primordia possess a significant increase in proliferating cells (*Aman et al., 2011*). Similarly, the BrdU index and the number of cells in 35 hpf NICD primordia is unchanged (*Figure 3G–I*).

The striking increase of the BrdU index in deposited *apc* neuromasts compared to the primordium suggests that *apc* neuromasts only grow larger once deposited, whereas the increase in the size of NICD neuromasts is controlled by a proliferation-independent process within the primordium, prior to deposition. To further test the hypothesis that aberrant proliferation does not contribute to increased neuromast size in NICD embryos, we inhibited proliferation by treating embryos with the DNA replication inhibitors hydroxyurea and aphidicolin (HUA and Aph, *Figure 3J–L*). Indeed, the size of the last deposited NICD neuromast is not significantly reduced, even though the primordium

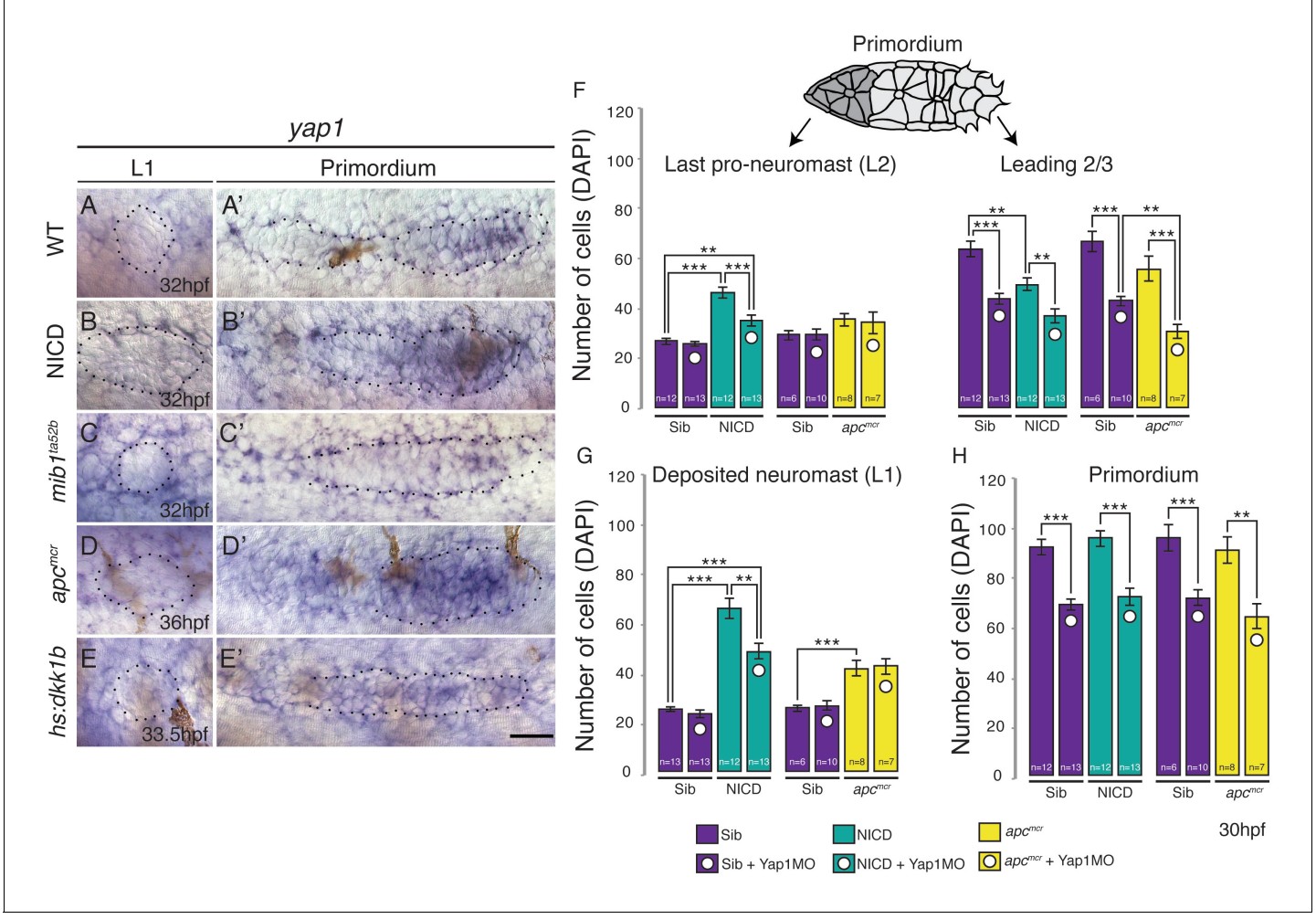

**Figure 2.** The Hippo pathway member *yap1* regulates primordium size but not neuromast size. (**A–E**) *yap1* is not expressed in WT neuromasts (**A**) or neuromasts in which Notch (**B** and **C**) or Wnt signaling (**D** and **E**) are manipulated. (**A'–E'**) *yap1* is upregulated in the primordium after (**B'**) Notch and (**D'**) Wnt overexpression compared to (**A'**) wildtype siblings and Wnt downregulation (**E'**). (**C'**) Notch loss in *mib1^ta52b* mutants leads to downregulation of *yap1*. (**F**) Loss of *yap1* by morpholino injections significantly reduces the number of cells in the leading 2/3 of WT, NICD and *apc^mcr* primordia, as well as the overall primordium size (**H**). (**G**) Notch and Wnt signaling significantly increases the number of cells in deposited L1 neuromasts, which is not rescued to a wildtype level by *yap1* morpholino injections. Even though the size of NICD L1 and L2 pro-neuromasts is significantly reduced in *yap1* morphants, they are still significantly larger than the corresponding WT neuromasts (**F** and **G**). Error bars represent standard error from one independent experiment (p<0.05=*, p<0.01=**, p<0.001=*** Student's *t* test). Scale bar is 25 µm.

The following figure supplement is available for figure 2:

**Figure supplement 1.** Hippo signaling components in the lateral line.

cell number is smaller (*Figure 3K–L*). On the other hand, *apc* neuromasts were rescued to a wildtype size and *apc* primordia became smaller (*Figure 3K–L*). To further test if Wnt and Notch affect neuromast size via independent mechanisms we simultaneously activated Wnt and Notch signaling. The induction of Wnt signaling in NICD embryos causes even bigger neuromasts than in untreated NICD embryos, suggesting an additive effect of the two pathways (*Figure 3M–O*). These results demonstrate that Wnt signaling affects neuromast size via upregulating proliferation in the deposited neuromasts, whereas activation of Notch causes an increase in neuromast size independently of the proliferation rate.

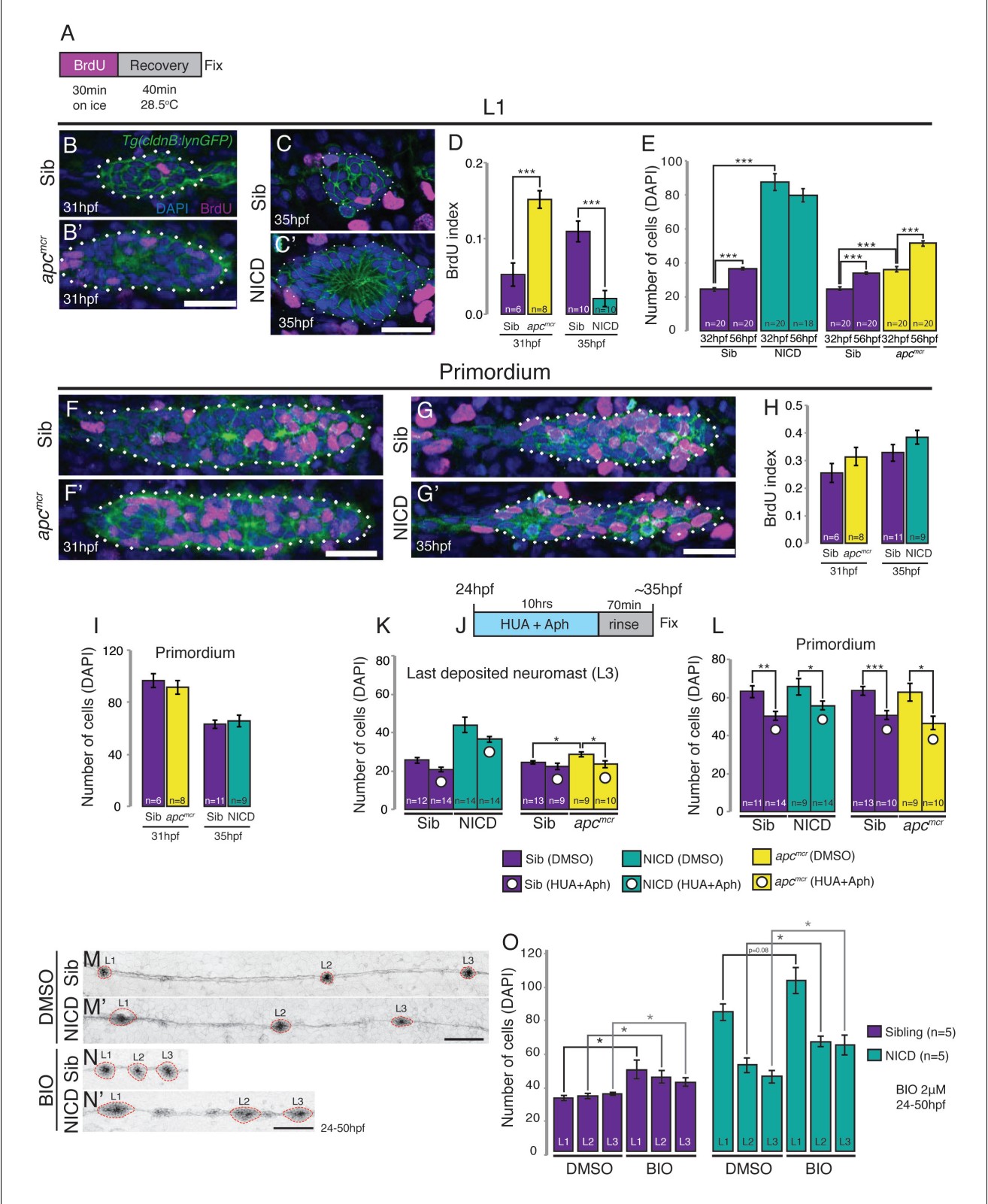

**Figure 3.** Proliferation is not responsible for large neuromasts in NICD embryos. (**A**) BrdU treatment strategy. (**B–C'**) Representative images of BrdU-positive nuclei in L1 *apc^mcr* (**B'**) and NICD neuromasts (**C'**). (**B** and **B'** and **D**) The BrdU index is increased in *apc^mcr* L1 neuromasts and significantly reduced in NICD L1 neuromasts (**C** and **C'** and **D**). (**E**) Nevertheless, the number of cells is significantly higher in the *apc^mcr* and NICD L1 neuromasts. While the *apc^mcr* neuromast grows overtime, the NICD neuromast size does not significantly change from 32 hpf to 56 hpf. (**F–G'**) Representative

*Figure 3 continued on next page*

Figure 3 continued

images of BrdU positive nuclei in *apc^mcr* (F') and (G') NICD primordia. (F–H) There is no significant difference in the BrdU index (H) or the number of cells (I) between sibling and *apc^mcr* or sibling and NICD primordia. (J) Hyroxyurea and aphidicolin treatment strategy. (K) Cell cycle inhibition with hydroxyurea and aphidicolin significantly reduces the number of DAPI- positive cells in the last deposited neuromasts in *apc^mcr* embryos. There is no significant reduction in sibling or NICD neuromast sizes after cell cycle inhibition. (L) Cell cycle inhibition significantly reduces the number of cells in sibling, *apc^mcr* and NICD primordia. Scale bars are 25 µm. Treatment of sibling and NICD embryos with the GSK3β inhibitor BIO causes significant enlargement of neuromasts (L1–3). Scale bar is 100 µm. Error bars represent standard error from one independent experiment (p<0.05=*, p<0.01=**, p<0.001=*** Student's *t* test).

## The increase in organ size is independent of the role of Notch in cell type specification

As proliferation does not contribute to the large proneuromast size in NICD primordia and Notch is crucial for cell fate specification via lateral inhibition, we asked if in NICD primordia a switch in cell fate could be responsible for the increase in organ size. As yet, four cell types have been identified in the primordium: (a) mesenchymal, unpatterned cells in the leading region, (b) hair cell precursors, (c) support cells in the center and (d) future interneuromast cells that are deposited in between neuromasts (*Figure 1A'*, arrows). Interneuromast cells are lateral line stem cells that postembryonically give rise to additional sensory organs (*Grant et al., 2005*; *López-Schier and Hudspeth, 2006*; *Lush and Piotrowski, 2014*). To test if Notch changes the fate of one cell population into another we counted the number of cells in these four populations in wildtype and NICD primordia.

In the ear and lateral line, Delta/Notch signaling is essential for the specification of sensory hair and support cells (*Haddon et al., 1999*; *Itoh and Chitnis, 2001*; *Riley et al., 1999*; *Millimaki et al., 2007*). The *deltad-* (and *atoh1a*) – expressing cell differentiates into a hair cell, whereas the surrounding Notch-expressing cells are specified as support cells (*Matsuda and Chitnis, 2010*). As Notch signaling inhibits *atoh1a*, NICD neuromasts consist only of support cells (*Figure 4A–L*). However, within the primordium only one *atoh1a*-labeled hair cell precursor exists per proneuromast (*Figure 4K*) and their cell fate switch to a support cell cannot account for an increase of an average of 15 or 11 cells in the first proneuromast (orange, *Figure 4M–N'*) and an increase of 12 or 19 cells in the last proneuromast cells at 26 hpf and 30 hpf (green, *Figure 4M–N'*). Nevertheless, we tested if the ectopic generation of hair cells in NICD neuromasts would rescue the neuromast size. We induced hair cells in NICD primordia and neuromasts by activating the hair cell specification gene *atoh1a* using a heat-shock inducible transgenic line (*Figure 4—figure supplement 1A–E''*). Hair cell containing neuromasts in heat-shocked *Tg(hs:atoh1a;NICD)* larvae are not significantly different in size from sibling NICD neuromasts (*Figure 4—figure supplement 1F*). Thus, the loss of hair cells in NICD neuromasts does not cause the increase in NICD neuromast size. Interestingly, *hs:atoh1a* alone induces a significant increase in neuromast size (*Figure 4—figure supplement 1C'' and F*). The increase of *her4* expression in response to ectopic *atoh1a* activation suggests that Atoh1a possibly induces larger neuromasts through Notch signaling activation in the primordium (*Figure 4—figure supplement 1B–E*).

We then asked if a cell fate switch in mesenchymal leading cells contributes to the large neuromast phenotype in NICD. We determined that the number of mesenchymal cells in the leading region is initially normal in 26 hpf embryos and is only reduced from an average of 9 to 5 cells in 30 hpf NICD primordia (*Figure 4M and N'*). Therefore, the reduction in the number of leading region cells is equally insufficient to explain the increase in organ size. The last potential cell fate switch that could contribute to the organ size increase is the loss of future interneuromast cells from the primordium. Future oblong-shaped interneuromast cells are located at the periphery of the primordium (*Figure 4M'*, arrow) but their number is not affected in NICD embryos (*Figure 4O–P* arrows, *Figure 4Q*). The only significant difference we detected between NICD and sibling primordia is that NICD primordia possess on average fewer, larger proneuromasts at the expense of a normal-sized proneuromast (*Figure 4M and M'*, grey cells, and *Figure 4R*). We therefore conclude that cell fate changes are not major contributors to the increase in organ size in NICD primordia but that the allocation of support cells into fewer but larger proneuromasts is the major cause.

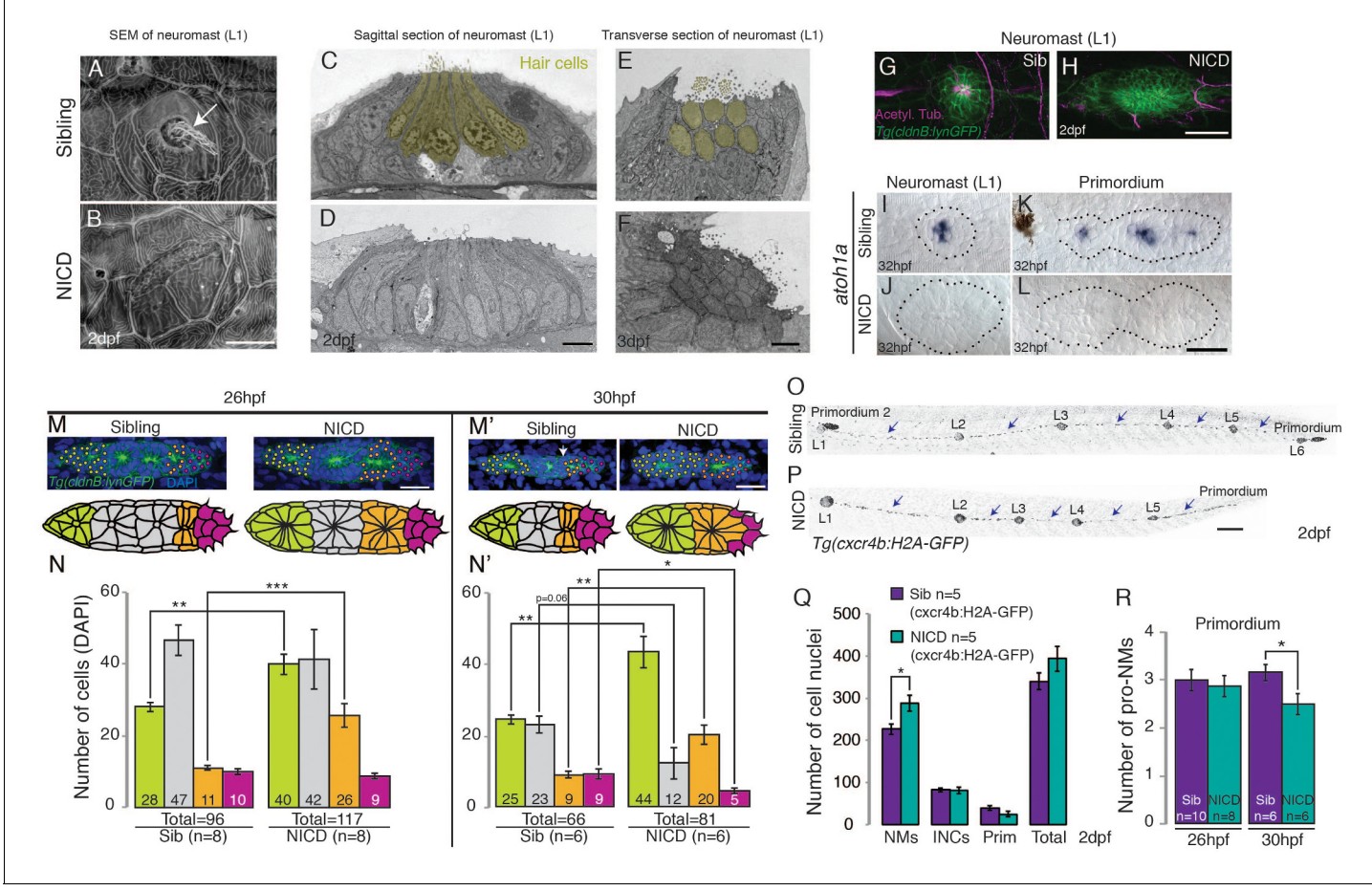

**Figure 4.** NICD neuromasts are not larger because of a switch in cell fate. Scanning electron micrograph of a 2 dpf sibling (**A**) and NICD neuromast (**B**) shows that no hair cells (white arrow in WT) are present in the NICD neuromast. Scale bar is 10 µm. (**C–F**) Transmission electron sections through sibling (**C,E**) and NICD neuromasts (**D,F**). (**C,E**) Hair cells are false colored in yellow. (**D,F**) No hair cells are present in NICD neuromasts. (**C–F**) Scale bars are 5 µm. (**G** and **H**) Acetylated tubulin antibody stains hair cells in a sibling neuromast (**G**), but staining is absent in a NICD neuromast (**H**). (**G** and **H**) Scale bar is 25 µm. (**I** and **L**) The proneural gene *atoh1a* is not expressed in NICD neuromasts (**J**) and the primordium (**L**). (**I–L**) Scale bar is 25 µm. (**M–N'**) Cell number (DAPI counts) in the different parts of the primordium. Magenta indicates mesenchymal tip cells, orange indicates the first proneuromast and green indicates the about to be deposited proneuromast. The gray cells are calculated by subtracting mesenchymal, first pro-neuromast and last pro-neuromast cell numbers from the total number of cells in the primordium. (**M** and **M'**) Scale bar equals 25 µm. (**O,P**) *Tg(cxcr4b:H2A-GFP)* labels all lateral line nuclei. (**Q**) NICD neuromasts consist of significantly more cells compared to sibling neuromasts, but there is no difference between the number of interneuromast cells (INCs) or the primordium cells at 2 dpf. (**O** and **P**) Scale bar equals 100 µm. (**R**) Significantly fewer proneuromasts are formed in a NICD primordium at 30 hpf. Error bars represent standard error from one independent experiment (p<0.05=*, p<0.01=**, p<0.001=*** Student's *t* test).

The following figure supplement is available for figure 4:

**Figure supplement 1.** Overexpression of *atoh1a* induces neuromast size through activation of Notch signaling.

## Notch is sufficient to induce organ morphogenesis and apical constrictions cell autonomously in the absence of Fgf signaling

The loss of Fgf signaling leads to the loss of proneuromasts/rosettes (*Lecaudey et al., 2008*; *Nechiporuk and Raible, 2008*). It has therefore been proposed that the Fgf-expressing central cell in a proneuromast acts as a signaling center that recruits surrounding support cells by inducing Fgf signaling in these cells (*Lecaudey et al., 2008*; *Nechiporuk and Raible, 2008*). We therefore tested if in NICD proneuromasts, Fgf ligand expression or Fgf signaling is increased. Surprisingly, even though *fgfr1* is normally expressed, *fgf3, fgf10a* and the Fgf targets *pea3 and dkk1b* are substantially

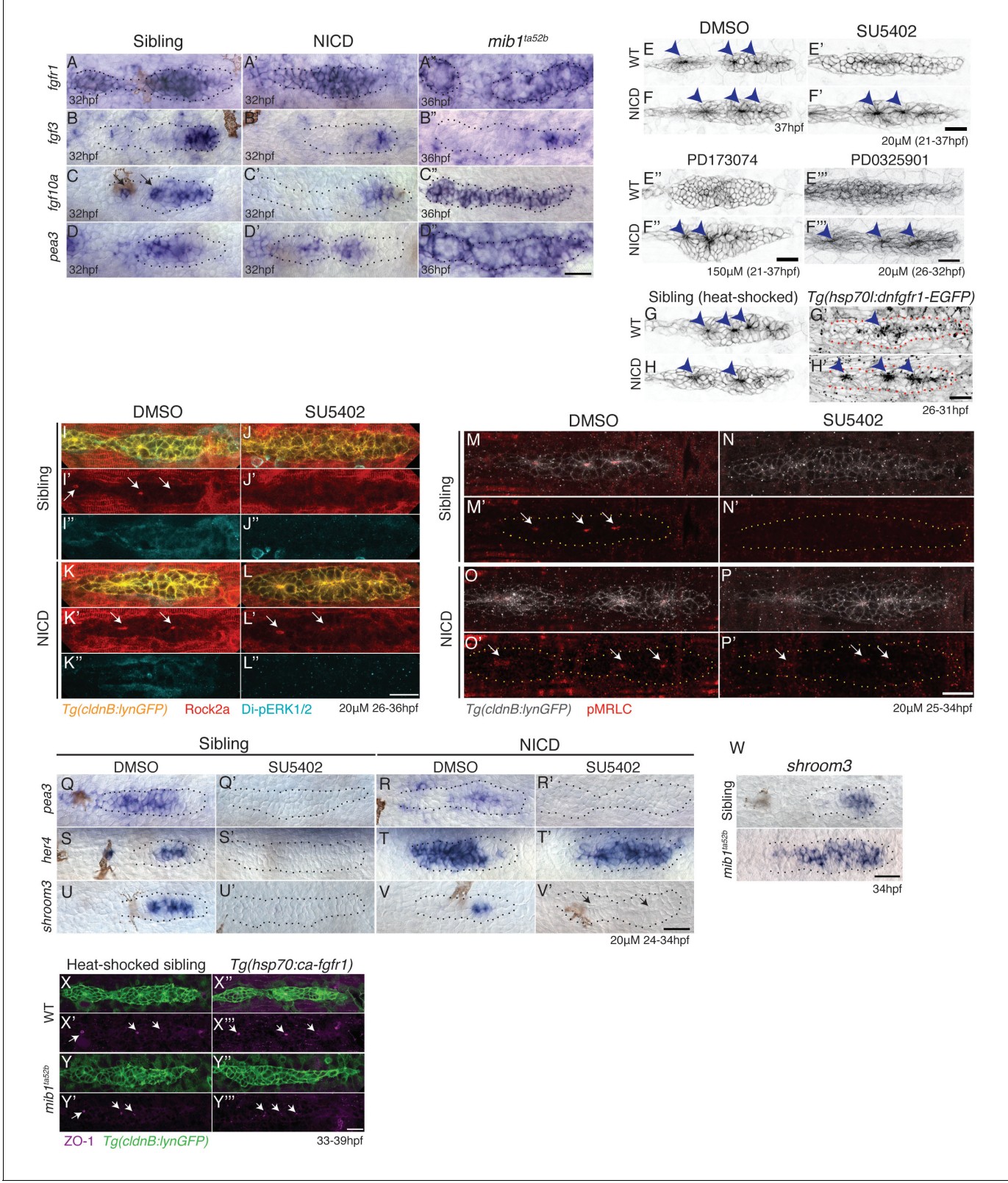

**Figure 5.** Notch induces proneuromast (rosette) formation independent of Fgf-MAPK-Shroom3 signaling. (A–D'') The expression of Fgf pathway members in the primordium is altered by Notch signaling. Fgf ligand (B' and C') and target gene expression (D') is largely downregulated in NICD primordia compared to siblings (B–D). (B'') *fgf3* and (D'') *pea3* expression is downregulated in *mib1^ta52b* primordia, but *fgf10* expression (C'') is largely expanded. *fgfr1* expression remains normal in NICD (A') but is downregulated in the center of the *mib1^ta52b* primordia (A''). (E–H') Notch induces

*Figure 5 continued on next page*

*Figure 5 continued*

proneuromast formation independent of Fgf-MAPK signaling. NICD primordia form proneuromasts in the presence of Fgfr1 and MAPK inhibitors (**F'** and **F''** and **F'''**) as well as after the induction of dominant-negative Fgfr1 (**H'**), while sibling proneuromast formation fails after these manipulations (**E'–E'''** and **G'**). (**G–H'**) Embryos were heat-shocked 2 times at 39°C for 20 min with 20 min incubation at a room temperature in between, starting at 26 hpf and fixed 4 hr later. (**I–L''**) Notch regulates apical Rock2a accumulation independent of Fgf signaling. Contrary to sibling primordia (**J'**), the Rock2a expression is maintained in the apices of the forming proneuromasts in NICD primordia treated with the Fgfr1 inhibitor SU5402 (**L'**). (**J''** and **L''**) Fgf downregulation is confirmed by the loss of di-pERK1/2 expression in the primordium. (**M–P'**) Notch regulates Phosphorylated Myosin Regulatory Light Chain (pMRLC) expression independent of Fgf signaling. (**N'**) Sibling primordia lose the pMRLC expression after Fgfr1 inhibitor treatment, but pMRLC expression is maintained in the apices of treated (**P'**) NICD proneuromasts. (**Q–V'**) NICD primordia form proneuromasts (black arrows) in the absence of *shroom3* expression (**V'**). Treatment with the Fgfr1 inhibitor SU5402 depletes *pea3* (**Q'** and **R'**), *her4* (**S'**) and *shroom3* (**U'** and **V'**) expression in control and NICD primordia. (**T'**) *her4* expression is maintained in NICD primordia after Fgfr1 inhibition. (**W**) *shroom3* is upregulated in *mib1^{ta52b}* primordia, although Fgf signaling is downregulated. (**X–Y'''**) Constitutive activation of Fgf signaling by heat-shock induction of Fgfr1 does not rescue rosette formation in *mib1^{ta52b}* primordia (**Y'''**) suggesting that Fgf signaling is not sufficient for rosette formation in the absence of Notch signaling. (**X–Y'''**) Embryos were heat-shocked at 39°C for 40 min, starting at 33 hpf and fixed 4 hr later. Scale bar in all panels is 25 μm.

The following figure supplement is available for figure 5:

**Figure supplement 1.** Notch signaling regulates proneuromast formation downstream of the MAPK pathway and shroom3 is dispensable for rosette formation.

abrogated in NICD primordia (***Figure 5A,A',B,B',C,C',D,D'***, Figure 8L,L'). Fgf signaling is likely reduced because: (a) Notch inhibits *atoh1a*, which is required for *fgf10a* expression in the most mature rosettes in the primordium (***Figures 4J,L*** and ***5C,C'*** arrows; [***Matsuda and Chitnis, 2010***]) and (b) because Notch inhibits Wnt, which normally induces Fgf signaling in the first forming rosette (see below, Figure 8I'–K'). Therefore, increased Notch signaling causes decreased Wnt and Atoh1a dependent Fgf signaling. The loss of Notch signaling in *mib1* mutants also leads to loss of Fgf signaling in central cells, however *fgfr1a* and *pea3* are strongly expressed in peripheral cells of the primordium (***Figure 5A'',D''***). In *mib1* primordia Fgf signaling is reduced in central cells because of the upregulation of *atoh1a*, which is normally restricted to a central cell by Notch signaling. Atoh1a, inhibits *fgfr1a*, thus causing the loss of Fgf signal transduction (***Figure 5A''–D''***; [***Matsuda and Chitnis, 2010***]).

As the expression of Fgf pathway genes is not completely abrogated in NICD primordia, we depleted Fgf signaling in NICD embryos with (a) the pharmacological Fgfr1 inhibitors SU5402 and PD173074 (***Figure 5E–F''***), (b) treated NICD embryos with an inhibitor of MAPKK/Di-pErk (PD0325901) (***Figure 5E'''–F''***, ***Figure 5—figure supplement 1A-B'***) and (c) heat-shock induction of dominant-negative *fgfr1* in NICD embryos (***Figure 5G–H'***). Irrespective of by which method we deplete Fgf signaling in NICD embryos, proneuromasts form, even though the primordium still eventually stalls (***Figure 5F',F'',F''',H'***). Also, the molecular analysis of Fgf-depleted wildtype and NICD embryos shows that, although the Fgf signal transduction protein di-pErk is absent from all embryos (***Figure 5J'', L''***), only in NICD embryos Rock2a and pMRLC are still properly localized to the apical constrictions of proneuromast cells (***Figure 5L',P'***). These results demonstrate that in the absence of Fgf signaling, Notch signaling is sufficient to apically localize Rock2a and pMRLC and induce/maintain apical constrictions (***Figure 5I-P'***).

To test if Notch signaling acts downstream of Fgf signaling we performed in situ analyses with the Notch target *her4* in Fgf-depleted sibling and NICD embryos (***Figure 5Q-T'***). Indeed, Notch signaling is highly reduced in the siblings in the absence of Fgf signaling (***Figure 5Q–S'***; [***Nechiporuk and Raible, 2008***]) but the loss of Fgf signaling has no effect on *her4* expression in NICD primordia (***Figure 5R–T'***). We determined that Notch acts downstream of the RAS/MAPK pathway, as the MAPKK inhibitor PD0325901 leads not only to the loss of the Fgf targets *pea3* and *fgfr1a* but also of *notch3* and *her4* (***Figure 5—figure supplement 1C–F'***).

To functionally test if Fgf signaling affects organ morphogenesis via the induction of Notch signaling, we crossed Notch-depleted *mib1* mutants with a heat-shock line that allows us to constitutively upregulate Fgf signaling (***Tg(hs:cafgfr1), Figure 5X–Y'''***). The analysis of the resulting embryos demonstrates that in the absence of Notch signaling, the activation of Fgf signaling is not sufficient to rescue proneuromast formation (***Figure 5Y–Y''***). Also, the expression of ZO1, a component of the

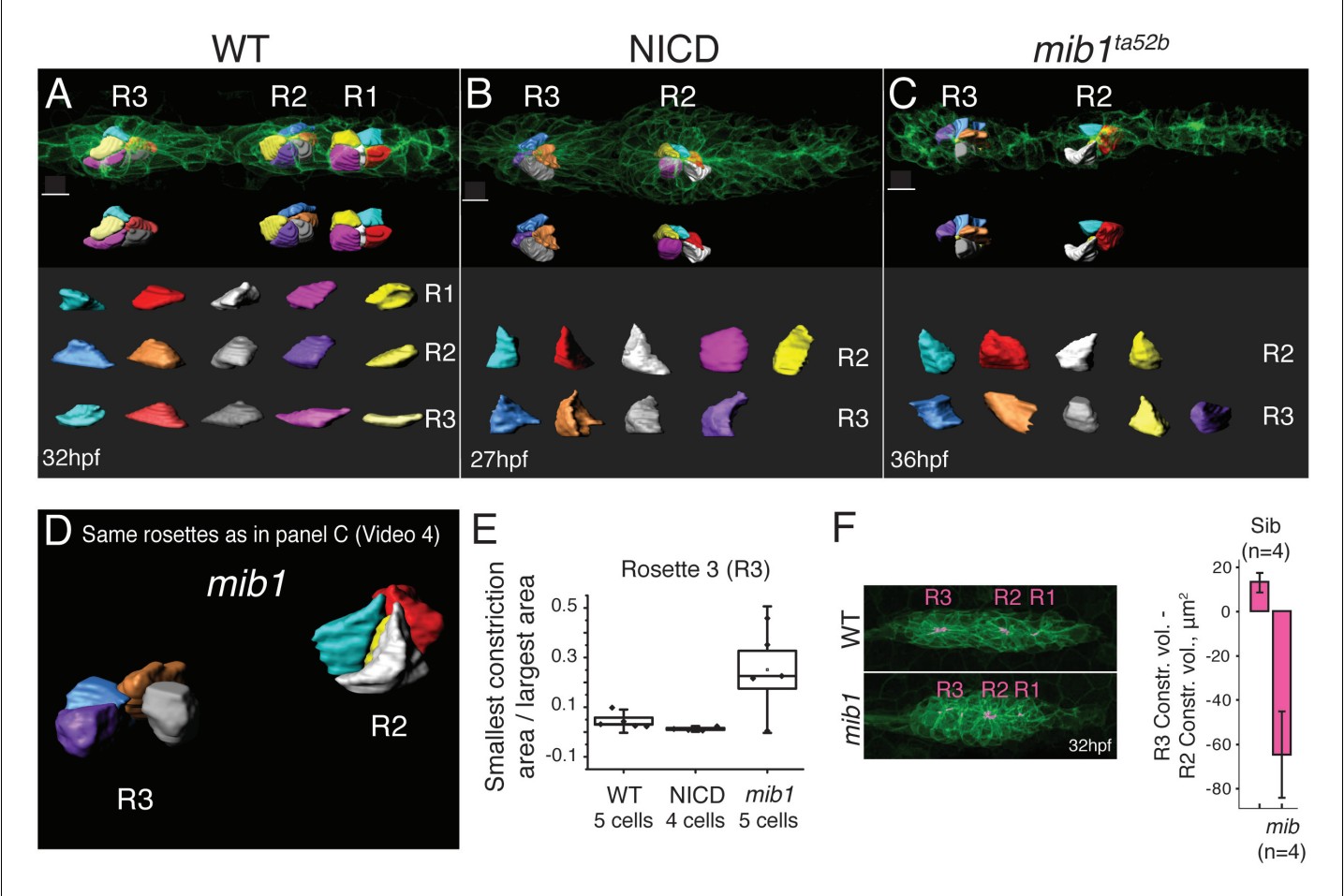

**Figure 6.** Loss of Notch signaling disrupts apical constrictions. (A–E) Analysis of cell shapes using Imaris software. R1–R3 indicate proneuromast numbers. (A) In wildtype and (B) NICD primordia all analyzed proneuromast cells in all rosettes constrict apically. Cells in a *mib1^ta52b* primordium (C) constrict in the more immature rosette (R2) but are lacking constricted apices in the last proneuromast (R3) at 36 hpf. The scale bars equal 10 μm. While the individual cells within one graph are scaled the same with respect to each other, no comparison can be made between different samples. This is due to the limitations of the Imaris scale bar function, rooted in the complexity of displaying 3D data. (D) Still image of an animation of the same primrodium as in (C) demonstrating the shape changes in R3. (E) Quantification of apical constrictions (area). Cells from the most mature proneuromasts (R3) were selected for analysis. Boxplot defines standard error and the ends of the whiskers show standard deviation between different cells. (F) The bar graph shows the apical constriction volume value when the R2 volumes are subtracted from R3 volumes. Constriction volumes gradually diminish in maturing proneuromasts in *mib1^ta52b* primordia, therefore, the last proneuromast (R3) in *mib* mutants has a significantly smaller constriction volume in comparison to R2, whereas, in the siblings R3 is larger than R2, which results in a positive value.

apically localized tight junctions is also still reduced in *mib1;hs:cafgfr1* embryos (**Figure 5Y',Y'''**) confirming that Notch acts downstream of Fgf signaling in rosette formation.

*shroom3* is an Fgf-dependent scaffolding protein thought to be required for preneuromast cell apical constriction (**Figure 5U**, [**Ernst et al., 2012**]). *shroom3* is lost in Fgf-depleted primordia, reduced in NICD primordia, completely lost in Fgf-depleted NICD primordia and upregulated in *mib1* primordia (**Figure 5U', V, V', W**). As NICD primordia are forming proneuromasts in the absence of *shroom3* (**Figure 5V'**, arrows), *shroom3* is not required for rosette formation in the presence of Notch. We also expressed *dominant negative shroom3* in wildtype primordia and did not observe any rosette or lateral line defects (**Figure 5—figure supplement 1G–I**). Likewise, in contrast to published data (**Durdu et al., 2014**; **Ernst et al., 2012**), *shroom3* morpholino injections did not delay rosette formation. To investigate if other *shroom* genes might act redundantly with *shroom3* we performed in situ hybridization with *shroom1, shroom2a, shroom2b* and *shroom4* (**Figure 5—**

*figure supplement 1J–K''*). *shroom2b* and *shroom4* are not expressed in the lateral line. *shroom1 and 2a* are not expressed in NICD primordia and we therefore conclude that *shroom3* does not act redundantly with other shrooms in NICD rosette formation.

Our results show that in wildtype primordia Fgfr activation induces Ras-MAPK signaling, which in turn activates Notch signaling. Notch leads to the apical localization of Rock2a and pMRLC in a *shroom3*-independent fashion, leading to apical constriction of the cell.

## A reduction in Notch leads to loss of cell-cell adhesion and apical constrictions

As Notch is sufficient to induce apical constrictions in the absence of Fgf signaling, we wondered if apical constriction is affected in *mib1* mutant primordia compared to wildtype and NICD primordia (*Figure 6*). Like in wildtype proneuromasts, cells in NICD proneuromasts constrict apically (*Figure 6A,B,E*). In 36 hpf *mib1* mutant primordia cells in younger proneuromast constrict normally (R2), however cells in the most trailing, about to be deposited rosette (R3) lose apical constrictions and no longer constrict apically (*Figure 6C–E*; *Video 4*). We measured cell constrictions in the R3 proneuromast by calculating the ratios between the narrowest cell surface area value by the widest area value. Results confirmed that *mib1* R3 cells are much less constricted than wildtype or NICD R3 cells (*Figure 6E*). We also visualized and quantified apical constriction volumes by coloring the constriction volume in magenta using Imaris (Bitplane) (*Figure 6F*). Subtracting the constriction volumes of R2 from R3 of WT and *mib1* proneuromasts reveals that the difference is positive for wildtype proneuromasts, whereas it is negative for *mib* proneuromasts. Therefore, the constriction volumes decrease in *mib1* as they mature and before they are deposited. These observations support the findings by Matsuda et al., that *mib1* proneuromasts fall apart (*Matsuda and Chitnis, 2010*). The disintegration of apical constrictions becomes more apparent between 32–36 hpf, as the loss of Notch signaling in *mib1* mutants becomes progressively more severe, likely because of depleted maternal stores (*Matsuda and Chitnis, 2010*).

## Notch-positive cells self-organize into rosettes

To test if NICD expressing cells might act as signaling centers that recruit surrounding wildtype cells into a proneuromast, we tested if the phenotype is non cell-autonomous. We transplanted cells (magenta color) from NICD transgenic embryos or wildtype embryos (as a control) into wildtype *Tg (cldnb:lynGFP)* embryos and compared neuromast size (green color) and cell composition between these two conditions (*Figure 7A–B'*).

We observed that neuromasts only get bigger (green dots) than the average wildtype neuromast (grey line) if they contain NICD clones but not if they contain wildtype clones (*Figure 7B*). If NICD cells acted as a signaling center enlarged neuromasts should not only contain NICD-positive cells but also possess an above average increase in wildtype cells (black dots). The quantification of all cell transplantation experiments confirmed that the significant increase in neuromast size in NICD>WT experiments (*Figure 7B'*, green bars) is only due to the excessive incorporation of transplanted NICD cells into proneuromasts, as the number of wildtype host cells (black bars) is not significantly changed between two types of cell transplantation strategies. Therefore, we conclude that wildtype cells are not recruited to contribute to larger neuromasts (*Figure 7A–B'*; *Video 5*). Thus, the increase in organ size after Notch activation is cell-autonomous and Notch transducing cells do not act as a signaling center.

### Notch signaling upregulates *e-cadherin* expression

It was suggested that self-organization into rosettes depends on cell adhesion molecules, such as *e-cadherin (cdh1)* and *n-cadherin (cdh2)*, and a reduction in *e-cadherin* expression correlates with lateral line adhesion defects in *mib1* mutants (*Matsuda and Chitnis, 2010*). The loss

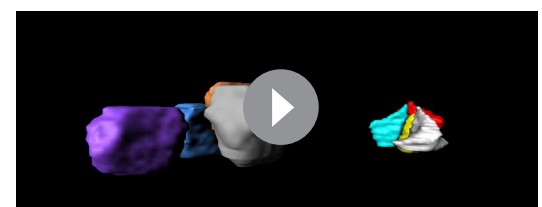

**Video 4.** Animation of the cells in the *mib1^ta52b* mutant primordium at 36 hpf (see *Figure 6C–D*). Cells constrict less in the trailing proneuromast (R3 in *Figure 6C–E*). Cell shapes are outlined and colored using Imaris (Bitplane) software.

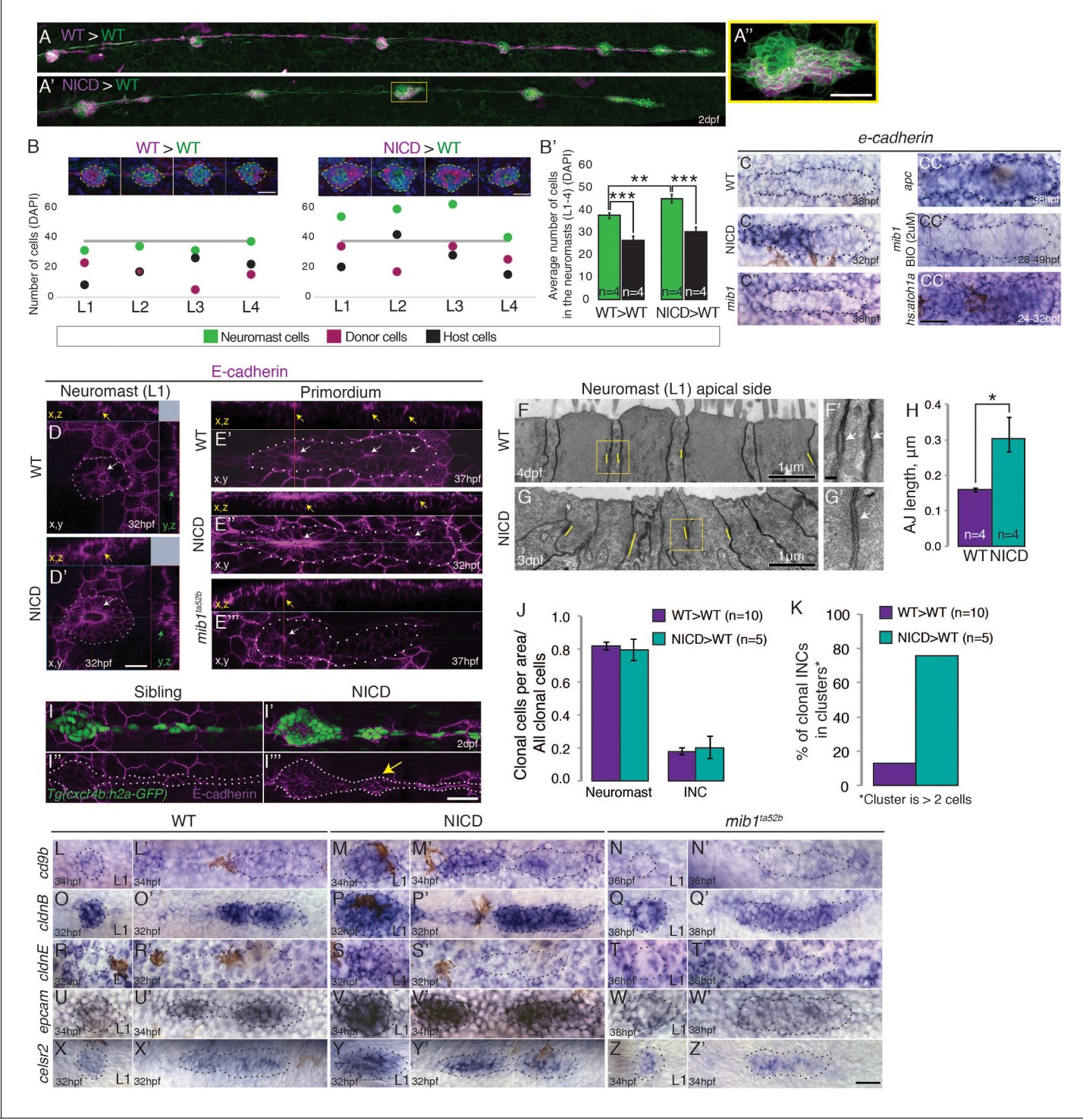

**Figure 7.** Notch cell-autonomously induces cell clustering. (**A,B** and **B'**) Wildtype cells transplanted into a wildtype embryo do not cause larger neuromasts. (**A'–B'**) Mosaic, NICD-positive cells that are transplanted into a wildtype embryo are clumping together and form bigger neuromasts. (**A''**) Mosaic L3 neuromast from panel (**A'**) where NICD positive cells cluster together to form an enlarged neuromast. (**B**) Images at the top of the graph show 2 examples of L1-L4 neuromasts with clone cell numbers measured below. Left: WT cells transplanted into WT lateral line and right: NICD cells transplanted into WT lateral line. Colored dots below indicate three values measured for each neuromast: green- total number of cells in the neuromast, magenta- number of transplanted cells, black- number of host cells (a calculated value of magenta cells subtracted from the number of green cells). The grey line indicates the average neuromast size of all WT neuromasts containing WT clones. (**B'**) Quantification of cell transplantation experiments. The significant increase in the neuromast size (green bars; 38 versus 45 cells) in NICD>WT experiments compared to WT>WT situation is

*Figure 7 continued on next page*

*Figure 7 continued*

only due to addition of transplanted NICD cells, since the number of host cells (black bar) is not significantly changed between two types of cell transplantation strategies (26 versus 30 cells). (**C–CC''**) *e-cadherin* expression in the primordium is upregulated by Notch (**C'**) and by Wnt signaling (**CC**) compared to a wildtype primordium (**C**). (**C''**) *e-cadherin* is downregulated in *mib1^ta52b^* primordia and in *mib1^ta52b^* in which Wnt is activated with BIO (**CC'**) suggesting that Wnt requires Notch signaling to induce *e-cadherin*. (**CC''**) *e-cadherin* is upregulated in *hs:atoh1a* primordia in which *her4* (Notch target) is also expanded (see ***Figure 4—figure supplement 1C***). (**D–I'''**) Notch upregulates E-cadherin protein expression in neuromasts (**D'**), proneuromasts in the primordium (**E''**) and in interneuromast (INCs) cells (yellow arrow) (**I''**) compared to a wildtype sibling (**D, E'** and **I,I''**). In *mib1^ta52b^* embryos *e-cadherin* is downregulated (**E'''**). Strongest E-cadherin expression is marked by the yellow arrows in the x,z plane, white arrows in the x,y plane and green arrows in the y,z plane. (**F–H**) NICD causes a significant increase in apical adherens junction (AJ) lengths. (**F'** and **G'**) Magnification of the areas in yellow boxes in **F** and **G**. AJs are marked by the white arrows. Error bar indicates standard error (p<0.05=* Student's *t* test). (**I–I''**) E-cadherin expression in the NICD interneuromast cells, which tend to form clusters (arrow), quantified in (**K**). (**J**) Transplanted NICD cells contribute in similar proportions to neuromasts and interneuromast cells as transplanted wildtype cells. (**K**) Transplanted NICD cells form significantly more clusters, as defined by groups of cells that contain two or more cells. (**L–Z'**) Apical junction genes, such as adherens junction genes and tight junction components are upregulated by NICD. (**L–N'**) *cd9b* is upregulated in NICD neuromasts and the primordium (**M** and **M'**) compared to siblings (**L** and **L'**), and is downregulated in *mib1^ta52b^* embryos (**N** and **N'**). (**O–Q'**) *cldnB* is upregulated in NICD neuromasts and the primordium (**P** and **P**) compared to a sibling (**O** and **O'**) and slightly reduced in *mib1^ta52b^* embryos (**Q** and **Q'**). The *cldnB* signal is especially low in the center of *mib1^ta52b^* neuromasts (**Q**). (**R–T'**) *cldnE* is upregulated in NICD neuromasts (**S**) compared to a sibling (**R**). *cldnE* is unchanged in *mib1^ta52b^* neuromasts (**T**). (**R', S'** and **T'**) No change in *cldnE* expression is seen in wildtype, NICD and *mib1^ta52b^* primordia. (**U–W'**) *epcam* is overexpressed in NICD neuromasts and the primordium (**V** and **V'**) compared to siblings (**U** and **U'**) but *epcam* is downregulated in *mib1^ta52b^* embryos (**W** and **W'**). (**X–Z'**) *celsr2* is overexpressed in NICD neuromasts and the primordium (**Y** and **Y'**) compared to siblings (**X** and **X'**) and *mib1^ta52b^* embryos (**Z** and **Z'**). All scale bars are 25 μm, unless stated otherwise. Error bars represent standard error (p<0.05=*, p<0.01=**, p<0.001=*** Student's *t* test).

The following source data and figure supplements are available for figure 7:

**Source data 1.** List of 187 genes that are upregulated in the lateral line of NICD embryos.
**Source data 2.** List of genes for cellular components from the GO term analysis in NICD primordium cells compared to a wildtype sibling.
**Figure supplement 1.** E-cadherin defficiency does not disrupt proneuromast formation.
**Figure supplement 2.** Upregulated GO terms for cellular components in NICD lateral line primordia.

of *e-cadherin* expression was attributed to the inhibitory effect of the expanded *atoh1a* domain in *mib1* primordia. Correspondingly, we found that in NICD primordia *e-cadherin* mRNA and E-cadherin protein are highly upregulated, accompanied by an increase in the length of adherens junctions in deposited neuromasts (***Figure 7C–C',D–E'',F–H***).

To determine if *e-cadherin* is regulated by Atoh1a, Wnt, Fgf or Notch signaling we analyzed *e-cadherin* expression after different signaling pathway manipulations (***Figure 7C–CC''***). Our data does not support the hypothesis that *atoh1a* negatively regulates *e-cadherin* and that therefore the upregulation of *e-cadherin* could be caused by the loss of *atoh1a* in NICD primordia. Ectopic expression of *atoh1a* in wildtype primordia causes increased *e-cadherin* levels and neuromast size (***Figure 7CC''*** and ***Figure 4—figure supplement 1F***). Simultaneously, *her4* is also induced (***Figure 4—figure supplement 1C and F***). These data suggest that heat-shock induction of *atoh1a* leads to the upregulation Notch signaling which induces *e-cadherin*.

Wnt overexpressing *apc* mutant primordia also show an increase in *e-cadherin* mRNA and protein expression suggesting that *e-cadherin* is regulated by Wnt signaling as in many other cell types (***Figure 7CC*** and ***Figure 7—figure supplement 1A,B***; ***Heuberger and Birchmeier, 2010***). Our data show that Wnt signaling regulates *e-cadherin* via the induction of Notch signaling, as *e-cadherin* is reduced in *mib1* primordia and the upregulation of Wnt with the GSK3β inhibitor BIO is not sufficient to induce *e-cadherin* in Notch-deficient primordia (***Figure 7C''* and 7CC'***). Likewise, Fgf signaling indirectly upregulates *e-cadherin* expression via the induction of Notch signaling. Loss of Fgf signaling leads to the loss of *e-cadherin* expression but if Notch is simultaneously activated, *e-cadherin* is expressed (***Figure 7—figure supplement 1C–F'***). *n-cadherin*, on the other hand, is regulated by Wnt in a Notch-independent fashion, as it is not affected in *mib1* or NICD primordia (***Matsuda and Chitnis, 2010***) but it is upregulated in *apc* mutants (***Figure 7—figure supplement 1G–J***). N-cadherin also localizes to apical constrictions (***Revenu et al., 2014***), however, we did not

**Video 5.** Time-lapse movie of a migrating *Tg(cldnB: lynGFP)* wildtype primordium into which magenta-colored NICD cells were transplanted. The NICD-positive cells contribute to a larger neuromast. The embryo was imaged every 5 min starting at 25 hpf.

observe a rosette defect in *n-cadherin* mutants (*pac^tm101b^*, *cdh2^hi3644^*, data not shown). Together these experiments show that *e-cadherin* is induced by Notch signaling downstream of Wnt, Fgf and Atoh1a.

To test the function of *e-cadherin* in rosette formation we transplanted cells from *e-cadherin* morpholino-injected embryos into wildtype embryos, as *e-cadherin* mutants die during epiboly (*Figure 7—figure supplement 1K–M''*; [*Kane et al., 2005*]). E-cadherin-deficient cells behave normally in wildtype primordia and neuromasts, even if almost the entire neuromast consists of mutant cells, suggesting that other adhesion molecules act redundantly (*Figure 7—figure supplement 1M–M''*; *Video 6*).

To determine if cell adhesion is affected at all in the NICD lateral line, we analyzed if transplanted NICD cells sort out from wildtype cells. First we analyzed if transplanted NICD cells preferentially contribute to neuromast or interneuromast cells and observed that NICD and wildtype cells contributed equally to either lateral line cell type, again demonstrating that cell fate is not affected (*Figure 7J*). However, NICD cells that contribute to interneuromast cells cluster significantly more often than wildtype cells, which are usually aligned in a string (*Figure 7I–I'''*, arrow and *Figure 7K*; *Figure 7—figure supplement 1N,O*). Almost 76% of NICD cells formed clusters of 2 or more cells, compared to 13% of wildtype cells. Combined with the finding that transplanted NICD cells contribute to wildtype neuromasts in larger clusters we conclude that Notch overexpression changes cell adhesion properties.

## Notch signaling upregulates cell-cell adhesion molecules and tight junction genes

To identify genes that are transcriptionally controlled by Notch signaling during proneuromast formation, we isolated primordium cells from dissected tails of 36 hpf *Tg(cldnB:lynGFP);Tg(cldnB:gal4) x Tg(UAS:nicd)* and sibling *Tg(cldnB:lynGFP);Tg(cldnB:gal4)* embryos by FACS and performed RNA-Seq analysis. We identified 187 genes that are upregulated in the lateral line of NICD embryos (*Figure 7—source data 1*). GO term analysis revealed that enriched cellular components are: 'apical junction complex', containing the tight junction components *cldnb*, *cldna*, *cldne* and *cingulinb* and 'actin cytoskeleton' (*Figure 7O–P' and R–S'*). Upregulated actin cytoskeleton genes are *formin1 (fmn1)*, *myo5c*, *baiap2b* and *cingulinb (cgnb)* (*Figure 7—figure supplement 2*, *Figure 7—source data 1* and *2*). *cgnb* is a tight-junction associated protein that links tight junctions to the actomyosin cytoskeleton and regulates RhoA signaling (*Aijaz et al., 2005*; *Terry et al., 2011*; *Van Itallie and Anderson, 2014*). Other molecules that interact with this complex are also upregulated, such as *rab25a*, a Rab11 GTPase family member, as is *celsr2,* an atypical cadherin (*Figure 7X–Y'*). Celsr1, if reduced, causes otic defects because of loss of apical constrictions due to disturbed actomyosin recruitment to the apical junctional complex (*Sai et al., 2014*). RNASeq analysis also revealed the induction of *cd9b* by Notch, a Tetraspanin family member implicated in cell-matrix adhesion and Sdf1 (Cxcl12a) mediated migration (*Figure 7L–M'*, [*Arnaud et al., 2015*; *Leung et al., 2011*]).

By performing in situ hybridization experiments with a number of candidate adhesion molecules, we identified that *epithelial cell adhesion molecule (epcam),* which regulates *claudin* expression and tight junctions is also strongly upregulated in NICD embryos (*Wu et al., 2013*; *Figure 7U–V'*). Conversely, in situ expression analyses demonstrated that molecules induced by NICD are downregulated in *mib1* mutants supporting the finding that Notch is regulating the expression of tight junction and adhesion molecules (*Figure 7N–Z'*).

**Video 6.** Time-lapse movie of a migrating *Tg(cldnB: lynGFP)* wildtype primordium into which magenta-colored E-cadherin morphant cells were transplanted. E-cadherin deficient cells integrate into the proneuromast normally and cause no aberrant rosette formation or maintenance phenotype. Embryo was imaged every 5 min starting at 25 hpf.

To determine if tight junction associated genes are responsible for the large organ phenotype we injected a *cldnb* morpholino into wildtype embryos, and also tested *celsr2*$^{rw71}$ mutants for lateral line phenotypes (*Kwong and Perry, 2013*; *Wada et al., 2006*). But like with *e-cadherin* mutant cells, we did not observe apical junction disassembly phenotypes, likely due to the presence of other *claudins* and *celsr1a/1b* in the primordium (data not shown). Likewise, a mutation in the adhesion molecule *epcam* does not cause a lateral line phenotype (*Slanchev et al., 2009*). *cd9b* has been previously knocked down by morpholino injections. Primordium migration is normal but neuromast formation is affected (*Gallardo et al., 2010*). As morpholino injections can be toxic (*Aman et al., 2011*), the function of *cd9b* is currently unclear and has to be reinvestigated.

Our data show that (a) cell adhesion is upregulated in NICD cells, as transplanted cells cluster and (b) cell adhesion molecules act redundantly as their individual downregulation does not cause a phenotype. Given that adhesion and tight junction genes belong to the most highly enriched GO term in NICD primordia, and NICD-positive cells form clusters, we conclude that Notch signaling induces an increase in organ size by activating a combination of adherence and tight junction molecules.

## Notch signaling is a component of the Wnt/Fgf signaling network that coordinates lateral line morphogenesis

Collective cell migration and organ morphogenesis are coordinated in the primordium via signaling interactions between Wnt and Fgf signaling (*Aman and Piotrowski, 2008*; *Chitnis et al., 2012*). It is therefore important to determine how Notch signaling fits into this gene interaction network. Wnt signaling in the leading primordium region activates Fgf signaling in the trailing region. Both pathways repress the other pathway via the activation of inhibitors in their respective expression domain (*dkk1b* and *sef/dusp6*).

Notch signaling is activated by Fgf signaling in the primordium (*Figure 5S'* and Figure 9A, B; [*Matsuda and Chitnis, 2010*; *Nechiporuk and Raible, 2008*]). Fgf signaling activates the transcription of *notch3* in hair cell progenitors and support cells and of *atoh1a* and *delta* ligands just in central hair cell progenitors (Figure 9B). Notch signaling, in turn, inhibits *atoh1* expression (*Baker et al., 1996*; *Baker and Yu, 1997*; *Matsuda and Chitnis, 2010*; *Millimaki et al., 2007*). However, if Notch signaling also acts downstream of Wnt signaling has not been investigated.

In *apc* mutants or in embryos treated with the GSK3β inhibitor BIO, the Fgf pathway genes *pea3* and *atoh1a* and the Notch target *her4* are upregulated (*Figure 8A–A', B-B', C-C'* and *Figure 8—figure supplement 1*). As Notch is activated by Fgf signaling we asked if Wnt activates Notch via the activation of Fgf signaling. We treated sibling and *apc* mutants with the Fgfr1 inhibitor PD173074. The loss of Fgf signaling attenuates *pea3, atoh1a* and *her4* in sibling and *apc* mutants, especially in the youngest proneuromast (*Figure 8D–F'*), which demonstrates that Wnt signaling is not sufficient to activate either *atoh1a* or *her4* to wildtype levels in the absence of Fgf (*Figure 8E,E' and F,F'*). Also, rosette formation still occurs in NICD primordia with depleted Wnt signaling by heat-shock activation of *dkk1b* demonstrating that Notch signaling acts downstream of Wnt signaling (*Figure 8G–H'*). In conclusion, Wnt activates Fgf signaling, which in turn activates *atoh1a*, *delta* ligands and Notch signaling (*Figure 9A,B*).

To test if Notch signaling negatively feeds back on Wnt signaling as in mature neuromasts (*Romero-Carvajal et al., 2015*), we assessed the expression of Wnt target genes in NICD and *mib1* mutant primordia. In NICD primordia, *lef1*, *wnt10a* and *sef* (an Fgf inhibitor that is regulated by Wnt; (*Aman and Piotrowski, 2008*)) are reduced, whereas in *mib1* the expression domains of these genes expand to the trailing region of the primordium (*Figure 8I-K'',P*). This finding was surprising as the Fgf target and Wnt inhibitor *dkk1b* is strongly reduced in NICD primordia and *wnt10a* should expand. *dkk1b* can therefore not be responsible for the repression of Wnt signaling in NICD primordia (*Figure 8L–L'*). A clue was provided by our observation that in *apc* primordia the Wnt target *wnt10a* is not expanded uniformly in the trailing region but is mostly upregulated in a group of central cells suggesting that Wnt signaling is inhibited in more peripheral cells (*Figure 8N,N'*, arrows). To test if Notch signaling restricts *wnt10a* expression in peripheral cells, we treated *apc* mutant embryos with the γ-secretase and Notch inhibitor DAPT. Indeed, *wnt10a* is now expressed in the most trailing cells demonstrating that Notch inhibits *wnt10a*, a feedback interaction not previously described in the primordium (*Figure 8O,O'*). A Fgf-independent inhibition of Wnt signaling by Notch is also supported by our finding that *wnt10a* expression is still inhibited in NICD primordia treated with Fgf inhibitor, even though loss of Fgf signaling in wildtype embryos leads to expanded

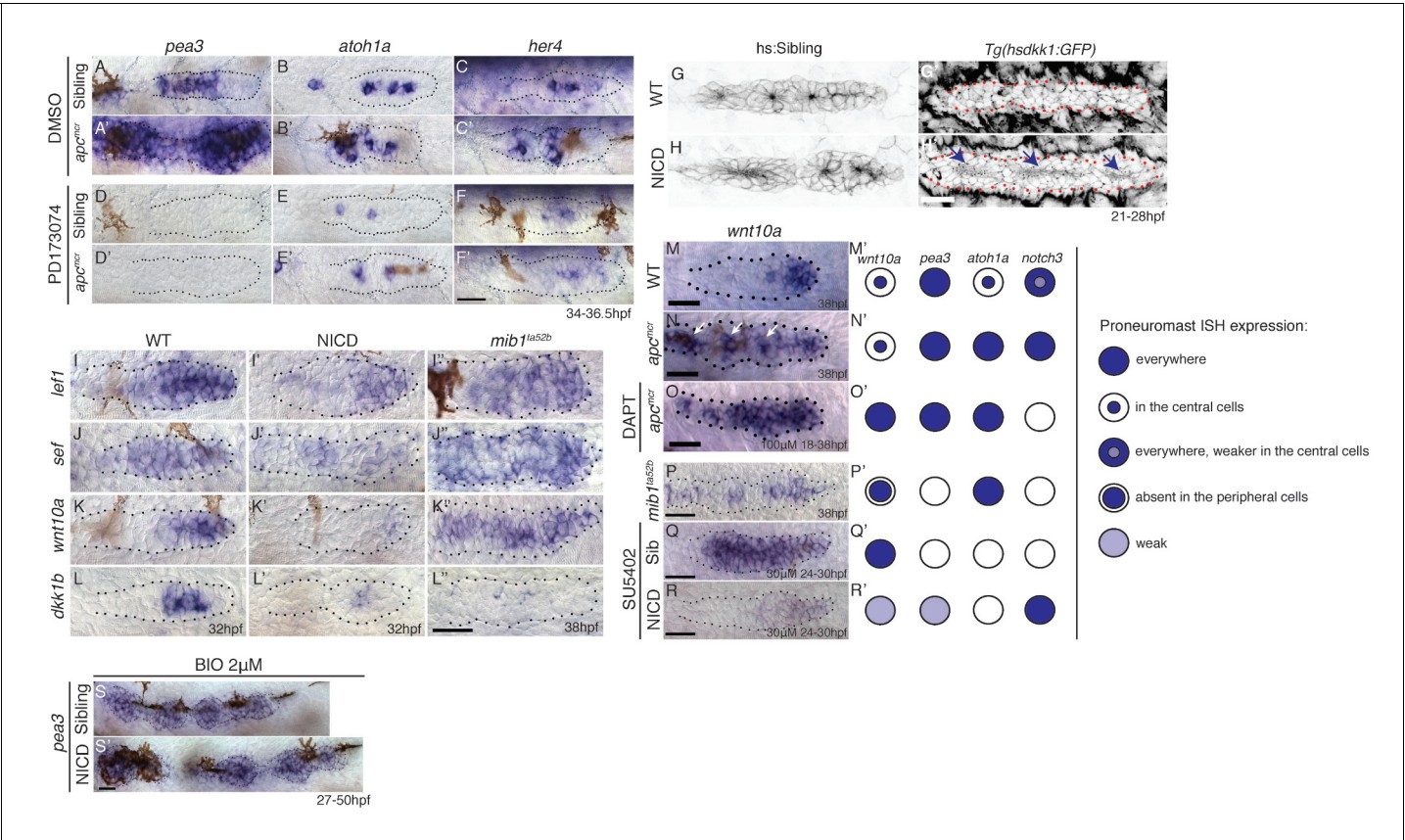

**Figure 8.** Wnt induces Notch signaling in the primordium via Fgf. (A and A') The expression of the Fgf target *pea3* is strongly increased in *apc^mcr^* primordia at 36.5 hpf. (B and B') In *apc^mcr^* primordia (B') *atoh1a* expression is increased compared to a sibling (B), but it still is restricted to individual cells. (C and C') *her4* expression is slightly expanded in the primordium after Wnt upregulation by *apc^mcr^* (C'). (D–F') The Fgf inhibitor PD173074 inhibits the expression of *pea3* (D') and prevents the upregulation of *atoh1a* (E'). The expression of *atoh1a* is lost in the first proneuromast, as *atoh1a* is Fgf dependent. The more trailing central cells express *atoh1a* because it becomes self-regulatory. The Fgf inhibitor downregulates *her4* in sibling (F) and *apc^mcr^* mutant primordia (F'). (G–H') Loss of Wnt signaling in the primordium by heat-shock induction of *dkk1b* does not disrupt proneuromast formation in NICD (H') compared to heat-shocked sibling primordia (G') suggesting that Notch acts downstream of Wnt and Fgf in proneuromast formation. (I–L'') Notch inhibits Wnt signaling in the primordium. The expression of Wnt targets *lef1* (I'), *sef* (J'), *wnt10a* (K') and *dkk1b* (L') expression is downregulated in NICD primordia. Conversely, in *mib1^ta52b^* primordia *lef1* (I''), *sef* (J'') and *wnt10a* (K'') are upregulated. (L'') *dkk1b* is downregulated in *mib1^ta52b^* primordia, because Fgf signaling is secondarily lost (see text). (M–R') *wnt10a* expression expands only in the absence of Notch signaling in the primordium. (M–N') In *apc^mcr^* primordia *wnt10a* expands towards the trailing region but is restricted to more central cells (arrows) (N,N'). (O,O') Downregulation of Notch with the γ-secretase inhibitor DAPT, causes a much more complete expansion of *wnt10a* in the trailing region of *apc^mcr^* primordia, demonstrating that Notch signaling inhibits *wnt10a* in the primordium. (Q–R') *wnt10a* expression is expanded in the absence of Notch signaling when primordia are treated with Fgf inhibitor (Q,Q') but is once again restricted to the leading region of the primordium if NICD is activated in the absence of Fgf (R,R'). (S) Treatment of sibling embryos with the GSK3β inhibitor and Wnt activator BIO causes the upregulation of *pea3*. (S') BIO treatment of NICD embryos rescues the loss of *pea3* in NICD primordia demonstrating that Wnt activates Fgf signaling upstream of Notch and that the loss of Fgf signaling in NICD is secondary to the loss of Wnt signaling. Wnt signaling is lost because Notch negatively feeds back to Wnt downstream of Fgf. All scale bars are 25 μm.

The following figure supplement is available for figure 8:

**Figure supplement 1.** Wnt signaling induces Notch signaling.

*wnt10a* expression (*Figure 8Q–R'*). The inhibition of *wnt10a* by Notch also explains why *wnt10a* is restricted to a single central cell in proneuromast of a wildtype primordium (*Figure 8M,M'*). *wnt10a* seems to be fluctuating in central cells, as it is not obvious in all primordia imaged (*Figure 8K*).

In conclusion, as illustrated by the schematic representation of gene expression in forming neuromasts, changes in the *wnt10a* domain only correlate with the presence or absence of Notch (*notch3*,

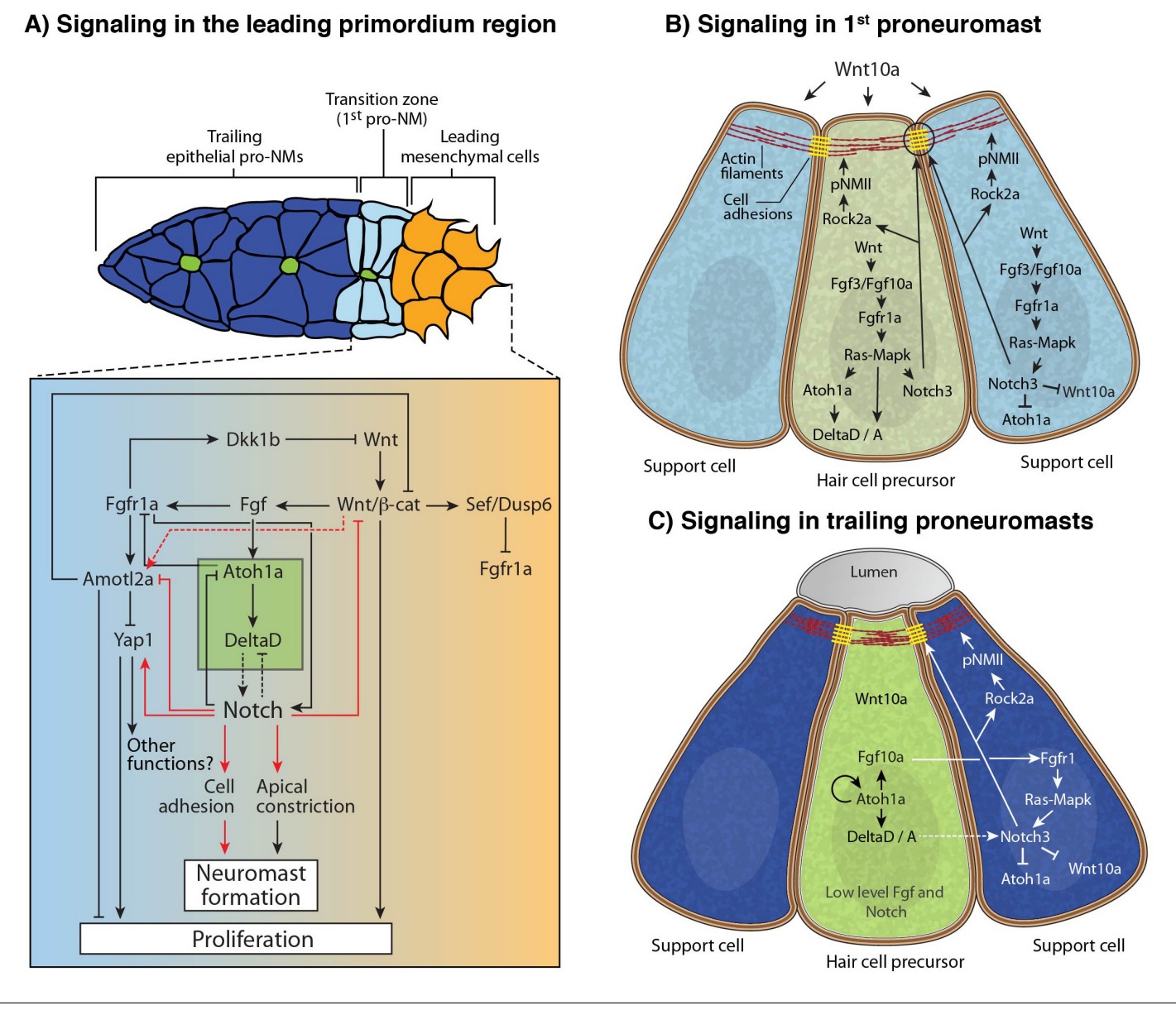

**Figure 9.** Model of the signaling interactions between Wnt, Fgf and Notch. (**A**) Signaling in the leading primordium region. Arrows in red indicate interactions described in this study. Other interactions are described in (*Agarwala et al., 2015*; *Chitnis et al., 2012*; *Matsuda and Chitnis, 2010*). Notch acts downstream of Wnt and Fgf signaling to form proneuromasts in the posterior lateral line via the upregulation of cell adhesion or/and cell apical constriction. Notch signaling also inhibits *amotl2a* and promotes *yap1* transcription, which induces some of the proliferation observed in the primordium. However, *yap1* does not affect neuromast size. Notch also negatively feeds back to restrict Wnt signaling, thus being an essential component of the signaling interactions that coordinate migration with organ formation. (**B**) Signaling in the most nascent proneuromast. Fgf signaling (*fgf3/10* and *fgfr1*) is induced by Wnt ligands in all proneuromast cells where it then induces *atoh1a* and Notch signaling (*Chitnis et al., 2012*) . As a result, Notch regulates apical constriction and cell adhesion in all proneuromast cells. (**C**) Signaling in the trailing proneuromasts. Fgf and Delta ligand expression now depends on Atoh1a and not Wnt anymore in central hair cell precursors (green). Fgf signaling is activated through Fgfr1 in the peripheral cells where it also induces Notch signaling (*Matsuda and Chitnis, 2010*). Notch signaling activates apical constriction and cell adhesion machinery in the support cells.

*Figure 8M'–R'*). Thus, the analysis of NICD primordia revealed that Wnt signaling is normally not only inhibited by the Fgf target *dkk1b* but also independently by Notch signaling.

The fact that Notch signaling negatively feeds back on Wnt signaling also explains why Fgf signaling is reduced in NICD primordia. Fgf signaling in the first forming proneuromast depends on Wnt

signaling, whereas the expression of *fgf10a* in the central cell of formed proneuromasts depends on *atoh1a* (*Aman and Piotrowski, 2008*; *Matsuda and Chitnis, 2010*). Therefore, the loss of Fgf signaling in NICD primordia can be attributed to both the inhibition of Wnt signaling and the inhibition of *atoh1a* expression by Notch in the trailing region. The conclusion that the loss of Fgf signaling in NICD primordia is mostly caused by Notch-mediated repression of Wnt signaling is confirmed by our finding that Fgf signaling (*pea3*) can be activated in NICD primordia by activating Wnt signaling with the GSK3$\beta$ inhibitor BIO (*Figure 8S,S′*). Combined our pathway analyses revealed that Notch is a component of the Wnt/Fgf signaling interaction network that controls migration with organ morphogenesis.

## Discussion

### Notch induces larger sensory organs independently of the Hippo pathway or proliferation

Even though overexpression of both Wnt and Notch signaling induce larger neuromasts, they produce this effect by different mechanisms. Wnt signaling upregulates proliferation in deposited neuromasts, whereas Notch signaling induces an increase in organ size as they form within the primordium in a proliferation-independent manner. Accordingly, cell cycle inhibition in NICD embryos does not rescue the neuromast size, in line with our previous study that demonstrated that proliferation does not influence wildtype neuromast size (*Aman et al., 2011*). Likewise, although inhibition of the Hippo pathway component *amotl2a* causes an increase in proliferation in the primordium (*Agarwala et al., 2015*) and *yap1* morpholino injections reduce the primordium size, neither manipulation affects the size of the forming sensory organs in wildtype, *apc* or NICD embryos (*Figure 2G–H*; [*Agarwala et al., 2015*]). Thus, primordium size and lateral line organ size are regulated by independent mechanisms. However, in *amotl2a* mutant embryos on average one more neuromast develops (*Agarwala et al., 2015*). Thus, primordium size does not affect neuromast size in a wildtype embryo but rather affects neuromast number. *yap1* is upregulated and *amotl2a* is reduced by Notch signaling without affecting the primordium cell number suggesting that *yap1* also possesses proliferation-independent functions that need to be further explored (*Figures 3H–I*, *9A*).

### Notch signaling upregulates components of the epithelial apical junctional complex

RNASeq and GO term analyses of FACS isolated primordium cells and the analyses of candidate genes revealed that members of the epithelial junctional complex are upregulated in the lateral line of NICD embryos. Epithelial junctional complexes consist of cadherin-based adherens junctions and tight junctions that encompass a large number of transmembrane proteins connected to the actomyosin belt via cytoplasmic scaffolding proteins (*González-Mariscal et al., 2003*). Because of the enrichment of cell adhesion molecules, such as E-cadherin and because transplanted NICD cells coalesce (*Figure 7K*, *Figure 7—figure supplement 1O*), we hypothesize that an increase in cell adhesion is contributing to the increase in organ size. However, transplanted E-cadherin mutant cells do not show a phenotype likely due to redundancy, as the primordium expresses several other adhesion molecules, such as *epcam* and *claudins*. Epcam is a calcium-independent, epithelial adhesion molecule and its reduction in mutant zebrafish and cultured cells changes their adhesiveness and induces ectopic, localization of apical junctional complexes (*Slanchev et al., 2009*; *Wu et al., 2013*). *epcam (tacstd)* is required for zebrafish gastrulation movements, and its loss also does not cause a lateral line phenotype suggesting that it acts redundantly with other adhesion molecules in the primordium. A previously reported morpholino-induced lateral line defect was likely caused by morpholino toxicity (*Villablanca et al., 2006*).

The upregulated Claudin proteins could act redundantly with E-cadherin and Epcam adhesion molecules. Although, Claudins are tight junction molecules and play an important role in controlling permeability of epithelia, they also affect cell adhesion as demonstrated in Xenopus development and cell culture (*Brizuela et al., 2001*; *Kubota et al., 1999*). In addition, their loss leads to epithelial to mesenchymal transition and the induction of epithelial characteristics if overexpressed (*Bhat et al., 2015*). Interestingly, loss of *claudin-6* leads to the loss of apical actin accumulation in the developing Xenopus pronephros demonstrating that Claudins regulate a variety of

morphogenetic processes (*Sun et al., 2015*). Thus, it is possible that Claudins are also involved in apical constriction, as has been demonstrated for the tight junction protein ZO-1 (*Bhat et al., 2015*; *Tornavaca et al., 2015*). The finding that the knockdown of individual adhesion and tight junction molecules does not cause a phenotype is reminiscent of the collectively migrating *Drosophila* border cells, which have only been shown to fall apart when JNK signaling is inhibited, which has many downstream targets (*Llense and Martín-Blanco, 2008*).

Other molecules previously implicated in proneuromast formation also act redundantly. For example, the combined downregulation of *lgl1* and *lgl2* leads to loss of the leading-most rosettes in the primordium (*Hava et al., 2009*). Lgl proteins act upstream of Notch signaling, as in the zebrafish retina loss of *lgl1* leads to an increase in the apical domain of neuroepithelia, which induces the upregulation of Notch signaling (*Clark et al., 2012*). Likewise, in the mouse brain *lgl1* regulates Notch via the localization of the Notch inhibitor Numb (*Klezovitch et al., 2004*). In NICD lateral line primordia *lgl2* expression is normal (data not shown) suggesting that *lgl2* acts upstream of Notch signaling in the lateral line as well. If *lgl1/2* are regulated by Fgf signaling has not been determined but *lgl1/2* could be a possible link between Fgf and Notch signaling, which needs further exploration.

## Shroom3 is not required for apical constriction of lateral line cells in the presence of Notch signaling

Apical constrictions are induced by myosin regulatory light chain activity that is phosphorylated by Rho associated coiled-coil protein kinase (ROCK; [*Aguilar-Cuenca et al., 2014*; *Bresnick, 1999*; *Sawyer et al., 2010*]). In the neural tube of mice and frogs the scaffolding and actin-binding protein *shroom3* is important for the apical localization of ROCK and neurulation (*Haigo et al., 2003*; *Hildebrand, 2005*; *Hildebrand and Soriano, 1999*; *Nishimura and Takeichi, 2008*). In contrast to previously published studies, our experiments did not reveal a function for *shroom3* in lateral line rosette formation (*Durdu et al., 2014*; *Ernst et al., 2012*). Most importantly, neuromast cells apically constrict normally in the absence of *shroom3* in NICD embryos (*Figure 5V–V'*, arrows). In chicken otocyst invagination also occurs independently of *shroom3* providing further evidence that apical constriction is achieved via different mechanisms in different contexts (*Martin and Goldstein, 2014*; *Sai et al., 2014*).

## Large organ size is not due to cell fate changes in NICD embryos

Notch signaling specifies support cells by inhibiting the proneural gene *atoh1a*, which is required for the acquisition of a neurogenic cell fate ([*Bermingham et al., 1999*], *Figure 4J,L* and *Figure 9C*). Accordingly, constitutive activation of Notch by overexpression of NICD leads to the loss of *atoh1a*-positive hair cell progenitors, whereas the loss of Notch signaling in *mib1* primordia causes an increase in *atoh1a* and *delta*-positive hair cell progenitors (*Figure 1L''–N''*, [*Itoh and Chitnis, 2001*]). Downregulation of *atoh1a* in *mib1* mutants modestly rescued *e-cadherin* expression and improved proneuromast cohesion, suggesting that *atoh1a* inhibits *e-cadherin* (*Matsuda and Chitnis, 2010*). A role for neurogenic genes in epithelial morphology was also described in *Drosophila*, where the loss of these genes causes delamination of cells from epithelia (*Hartenstein et al., 1992*). Nevertheless, it is unlikely that the NICD phenotype is caused by the loss of the central hair cell precursor and loss of *atoh1a* expression because: (a) re-expression of Atoh1a in NICD primordia does not rescue neuromast size (*Figure 4—figure supplement 1E'',F*) (b) *apc* and *hs:atoh1a* primordia show an increase in *e-cadherin*, even though *atoh1a* is robustly expressed, arguing that *atoh1a* does not inhibit *e-cadherin* expression (*Figures 7CC, 8CC''*; *Figure 7—figure supplement 1A,B*), (c) the loss of *atoh1a/b* by morpholino injection into wildtype embryos does not affect *e-cadherin* expression and does not lead to larger neuromasts (*Matsuda and Chitnis, 2010*; *Nechiporuk and Raible, 2008*). On the contrary, our data show that ectopic *atoh1a* causes a significant increase in neuromast size. Interestingly, Atoh1a also upregulates Notch target *her4* expression (*Figure 4—figure supplement 1B,C,F*). This suggests that neuromast size in *hs:atoh1a* embryos is possibly regulated by Notch signaling acting downstream of Atoh1a dependent Delta ligand expression. During zebrafish inner ear development Atoh1b is required for the initial activation of *her4* (*Radosevic et al., 2014*). Whether Atoh1a regulates Notch in the lateral line still needs to be further investigated.

The above experiments show that, even though NICD primordia loose hair cells, which could lead to signaling changes in neighboring cells, the loss of hair cells is not the cause of organ size increase of NICD proneuromasts.

## Signaling in forming and mature proneuromasts

Interestingly, during proneuromast formation and maturation signaling between central hair cell precursors and surrounding support cells changes slightly between the first forming proneuromast and the trailing, more mature proneuromasts (*Figure 9B and C*; [*Matsuda and Chitnis, 2010*]). In the first forming proneuromast *fgf3/10a* is expressed in all cells, whereas in trailing, more mature rosettes these ligands are restricted to a central cell. This restriction is controlled by Notch signaling that inhibits the proneural gene *atoh1a* in support cells (*Figure 9C*). In contrast, even though Fgf signaling is active in all cells in the first forming proneuromast, the downstream targets *atoh1a* and *deltaa/b* are immediately restricted to a central hair cell precursor. Likewise, *wnt10a* is only expressed in one central cell. The restriction of *atoh1a* to a central cell contrasts findings in flies where *atoh1* is first broadly expressed, marking a prosensory field and only subsequently becomes restricted during cell specification (*Jarman et al., 1995*; *Millimaki et al., 2007*). The mechanisms by which Notch signaling immediately inhibits *atoh1a*, *delta* and *wnt10a* expression in support cells and confines them to a central cell are still unknown. Another difference between signaling in the first forming and trailing proneuromasts has previously been described. Fgf signaling is induced in all cells of the first forming proneuromast by Wnt ligands, possibly *wnt10a* (*Figures 8K*, *9A,B*). In more trailing proneuromasts, Fgf signaling depends on *atoh1a*, rather than Wnt signaling (*Figure 9C*, [*Matsuda and Chitnis, 2010*]). Likewise, *atoh1a* expression becomes Fgf-independent and self-regulatory (*Matsuda and Chitnis, 2010*; *Millimaki et al., 2007*). Even though, Fgf and Notch signaling are particularly high in support cells because of lateral inhibition, some Fgf and Notch signaling occurs even in central cells, as evidenced by *her4* expression (*Figures 1J* and *5S*, not shown in *Figure 9C*), as well as *notch3*, albeit at lower levels (*Figure 1I* and *Figure 5—figure supplement 1E*). Possibly, this low level of Notch signaling ensures that also central cells maintain their apical constrictions.

## Notch cell-autonomously induces apical constriction and rosette formation downstream of Fgf signaling

The expression of Fgf ligands in a central cell in a forming neuromast suggests that this cell acts as a signaling center that organizes neighboring cells into a rosette (*Harding and Nechiporuk, 2012*; *Lecaudey et al., 2008*; *Nechiporuk and Raible, 2008*). Indeed, loss of Fgf signaling causes existing neuromasts to fall apart and a failure of new neuromasts to form (*Lecaudey et al., 2008*; *Lush and Piotrowski, 2014*; *Nechiporuk and Raible, 2008*). Durdu et al., reported that Fgf signaling in trailing proneuromasts depends on Fgf ligands being secreted into a dorsal lumen and that only central cells that are apically in contact with the lumen activate the Fgf target *pea3* (*Durdu et al., 2014*). They concluded that Fgf signaling and/or Fgf-dependent rosette formation is important for cell migration and the periodicity of organ deposition (*Durdu et al., 2014*; *Lecaudey et al., 2008*). However, lumina only form in the trailing region of the primordium and are not required for Fgf-mediated formation of the first proneuromast (*Figure 9B*). Our results demonstrate that Fgf signaling is dispensable for rosette formation in the presence of ubiquitous Notch signaling. This finding begs the question why *fgf10a* production is restricted to a central cell that limits the number of Fgf-responding and Notch-activating cells. A likely explanation is that restricting Fgf signaling and thereby Notch signaling is important to control organ size.

An interesting question is why in NICD primordia two larger rosettes form, rather than three smaller ones as in their siblings. We hypothesize that the Notch-driven increases in cell adhesion and cortical tension change the morphology of the organ to achieve minimal surface tension (*Heisenberg and Bellaïche, 2013*). Measurements and manipulations of physical forces are needed to test if physical forces determine organ size control and the number of organs that can self-organize in a defined tissue such as the lateral line primordium.

## Summary

Together, our experiments demonstrate that Notch signaling is an essential part of the feedback loop between Wnt and Fgf signaling that coordinates cell migration with sensory cell specification

and organ morphogenesis. Importantly, we show that Notch signaling acts cell-autonomously and downstream of Fgf signaling in sensory organ rosette formation via the regulation of cell adhesion and tight junction molecules. Therefore, Notch is involved in neuromast rosette self-organization that is analogous to rosettes and that precedes lumen formation in pre-implantation mouse embryos (*Bedzhov and Zernicka-Goetz, 2014*). Similarly, rosettes self-organize during *C. elegans* gastrulation but are also characteristic for some brain tumors (*Pohl et al., 2012*; *Wippold and Perry, 2006*). We speculate that Notch signaling might be involved in these other developmental events as well. Therefore, our results do not only inform lateral line biology but also contribute to our understanding of morphogenesis of other organs and tissues and how rosettes form in certain brain tumors.

## Materials and methods

### Fish maintenance and fish strains

The following fish strains were used

*Tg(cldnb:lynGFP)[zf106]* (*Haas and Gilmour, 2006*), *Tg(Tp1bglob:eGFP)[um13]* (*Parsons et al., 2009*), *Tg (UAS:myc-Notch1a-intra)[kca3]* (*Scheer and Campos-Ortega, 1999*), *Tg(hsp70l:Gal4-VP16)[VU22]* (*Shin et al., 2007*), *apc[mcr]* (*Hurlstone et al., 2003*), *mindbomb, mib1[ta52b]* (*Itoh and Chitnis, 2001*), *Tg(hsp70l:dkk1b-GFP)[w32]* (*Stoick-Cooper et al., 2007*), *Tg(hsp70l:dnfgfr1-EGFP)[pd1]* (*Lee et al., 2005*), *Tg(hsp70:ca-fgfr1)[pd3]* (*Marques et al., 2008*), *Tg(hsp70:atoh1a)[x20]* (*Millimaki et al., 2010*), *Tg(UASDNshroom3)* (created by injecting plasmid: pT2dest(bidirectional UAS)-dsred-shrm3DN(520-874)) received from B. Link (*Clark et al., 2012*; *Kwan et al., 2007*), *Tg(ubi:Zebrabow)* (*Pan et al., 2013*), *p53[zdf1]*, *pac[tm101b]*, *cdh2[hi3644]* kind gift from Anand Chandrasekhar. To generate the *Tg (−8.0cldnB:gal4vp16)[psi8]*, we used the zebrafish Tol2 kit (*Kwan et al., 2007*). The 8 kb claudinB promoter fragment (*Haas and Gilmour, 2006*) was cloned into the 5' entry vector and combined via a Gateway reaction with the Gal4-VP16 middle entry vector of the Tol2 kit. The DNA was injected into one-cell stage zebrafish embryos to generate transgenic fish. To generate the *TgBAC(cxcr4b:H2A-GFP)p2* transgenic line, the BAC clone DKEY-169F10 was modified in two ways by recombineering. First, the *Tol2* sites and the *cryaa:dsRed* transgenesis marker were inserted into the BAC backbone (*Fuentes et al., 2016*). Second, a cassette consisting of H2A-GFP flanked by 791 bp and 1042 bp of homology upstream of *cxcr4b* exon 2, and downstream of the *cxcr4b* stop codon, respectively, was inserted to replace the *cxcr4b* coding sequence in *cxcr4b* exon 2 (amino acid 6–358, the last amino acid before the stop codon) using seamless galK-mediated recombineering (*Warming et al., 2005*). This transgene expresses the first five amino acids from *cxcr4b* exon 1 fused to H2A-GFP from the *cxcr4b* promoter. The final BAC transgene was characterized by SpeI and EcoRI restriction digestion, sequencing of PCR amplicons of the modified locus, and BAC-end sequencing. The DKEY-169F10 BAC library were obtained from ImaGenes GmbH, Germany. BAC transgenes were purified with the nucleobond BAC 100 kit (Clontech). We co-injected 1 nl of 50–250 ng/ml BAC transgene DNA, and 40 ng/ml *Tol2* mRNA into the lifting cell of the zygote of 0- to 20-minute-old embryos. The *Tol2* mRNA was transcribed from pCS2FA-transposase (*Kwan et al., 2007*) using the mMessage Machine SP6 Transcription Kit (Thermo Fisher). At 4 days post fertilization (dpf), the rate of mosaic expression of the fluorescent protein in the lens was scored. Clutches with 50% or more embryos showing mosaic fluorescent protein expression in the lens were raised to adulthood. Stable transgenic larvae were identified by out-crossing adults injected with the *cxcr4b:H2A-GFP* BAC transgene, and by raising larvae positive for the fluorescent transgenesis marker in the lens at 4 dpf. We screened 100 or more embryos from eight adults injected with the *cxcr4b:H2A-GFP* BAC transgene for fluorescent transgenesis marker expression in the lens and identified three founder fish. The full names of the transgenic lines identified are TgBAC(*cxcr4b:H2A-GFP; cryaa:dsRed*) p1 through p3. All experiments were performed according to guidelines established by the Stowers Institute IACUC review board.

### In situ hybridization

In situ hybridization procedures were performed as described (*Kopinke et al., 2006*). The following probes were used: *lef1, pea3, sef, axin2, fgf3, fgf10, fgfr1, dkk1b* (*Aman and Piotrowski, 2008*), *wnt10a* (*Lush and Piotrowski, 2014*), *deltaa, deltab, deltac, deltad* (*Itoh and Chitnis, 2001*), *dGFP, notch3, her4.1, atoh1a* (*Jiang et al., 2014*), *s100t* (*Venero Galanternik et al., 2015*), *shroom3* (*Ernst et al., 2012*), *epcam* (probe cb6 from ZIRC), *celsr2* (*Wada et al., 2006*; *Siddiqui et al., 2010*),

*cldne* kind gift from Ashley Bruce. The following primers were used to clone other probes: *yap1* (Fw 5'-CGACTTTCCTTGAAAACGGT-3' and Rv 5'-AAGGTGTAGTGCTGGGTTCG-3'), *stk3* (Fw 5'-GCAG TGCTTCCTTAAACTCCAAAC-3' and Rv 5'-GCAGGAATCTAGAGTAAGATGCAG-3'), *amotl2a* (Fw 5'-TGGAGAAGGTGGAAAGGATG-3' and Rv 5'-GCTGGGCTCTTCTGAATCAC-3'), *shroom1* (Fw 5'-CGTCTATGATGGGCAAACCT-3' and Rv 5'-GGCAGGTCGTATGAGATGGT-3'), *shroom2a* (Fw 5'-AACAAGCAAACCCAATGGAG-3' and Rv 5'-CTTTGGAGGCGAGTTGTAGC-3'), *cdh1* (Fw 5'-TGGAAGAACAAGGACGCTCT-3' and Rv 5'-TCTCAGGGACAGATGCAGTG-3'). The following probes were cloned into pPR-T4P vector using AA18 and PR244 primers: *cldnb* (Fw 5'-CATTACCA TCCCGAAACGAAAAGCATGGCATC-3' and Rv 5'- CCAATTCTACCCGAGAGGCTGTTTCAAACG TGG-3'), *cd9b* (Fw 5'-CATTACCATCCCGTTGTGTTCACACACTCGCTG-3' and Rv 5'- CCAATTC TACCCGACAACAGGACAACCACTCGC-3').

## Genotyping

The following primers were used to identify *Tg(hsp70:ca-fgfr1)^{pd3}* embryos by PCR: Fw 5'- GCAGCC TGACAGGACTTTTC-3' and Rv 5'-GATCCGACAGGTCCTTTTCA-3'.

## Transplantation assay

Transplantation assays were performed as previously described (*Aman and Piotrowski, 2008*). Donor embryos were injected with 3% lysine-fixable biotinylated-dextran AF568 (D22912) (Invitrogen, USA) at the one cell stage. Alternatively, *Tg(ubi:Zebrabow)* embryos were used as donors in *Figure 7*.

## Morpholinos

All embryos were injected at the one cell stage. The splice-blocking *yap1* morpholino (5'-AGCAACA TTAACAACTCACTTTAGG-3') was injected at a concentration of 10 ng/2 nL (*Skouloudaki et al., 2009*). Translation blocking E-cadherin morpholino (5'-ATCCCACAGTTGTTACACAAGCCAT-3') was injected at a concentration of 8.4 ng/2 nL (*Babb and Marrs, 2004*).

## Immunohistochemistry and Phalloidin staining

Embryos for Phalloidin staining were fixed for 30 min in 4% paraformaldehyde (PFA), washed 3 times for 5 min with 0.8% TritonX in PBS (PBT) and incubated for 2 hr at room temperature with Phalloidin (Invitrogen, 1:40 in 2% PBT). For immunohistochemistry embryos were fixed in 4%PFA at 4°C for the following antibodies: c-Myc (9E10) Santa Cruz Biotechnology (sc-40) (1:500), Anti-Acetylated Tubulin (T679) Sigma-Aldrich (1:500), mouse anti-ZO1 (Zymed; 61–7300) (1:200). Glyo-Fixx (Thermo Scientific, UK) at 4°C was used as a fixative for these antibodies: Di-pERK1/2 (Diphosphorylated ERK-1 and 2 antibody-mouse (M8159) Sigma-Aldrich) (1:500), Anti – ROCK – 2a (CT), Z – FISH- rabbit (AS-55431) (1:50), Mouse Anti-E-Cadherin (610182) BD-Biosciences (1:500), and Bouin's fixative (Polysciences) was used for Phospho-Myosin Light Chain 2 (Ser19) Antibody (#3671 Cell Signaling (1:20)).

## Electron microscopy

### Transmission electron microscopy (TEM)

Samples were fixed in 2.5% glutaraldehyde and 2% paraformaldehyde in PBS for 1 hr at room temperature (then stored in 4°C until processing), followed by a secondary fixation in 1% aqueous osmium tetroxide with potassium ferricyanide overnight at 4°C. Samples were then dehydrated in a graded series of ethanol with propylene oxide as a transitional solvent, and infiltrated in Epon. Sections were cut on a Leica UC6 ultramicrotome in the range of 50 nm to 70 nm thickness and post stained with 5% uranyl acetate in 70% methanol and Sato's lead stain. Sections were viewed in a FEI Technai Spirit BioTWIN TEM and imaged with a Gatan Ultrascan digital camera.

### Scanning electron microscopy (SEM)

Samples were fixed in 2.5% glutaraldehyde and 2% paraformaldehyde in PBS for 1 hr at room temperature (then stored at 4°C until processing) and treated with tannic acid, osmium, thiocarbohydrazide, and osmium (TOTO) as described in *Jongebloed et al. (1999)*. Samples were then dehydrated

in an ethanol dilution series, critical point dried in a Tousimis Samdri 795 CPD, mounted on stubs and coated with gold palladium. Samples were imaged in a Hitachi TM-1000 tabletop SEM.

## Heat-Shock treatments

Heat-shock induction was done at various developmental time points, different temperature combinations and time intervals as indicated in the text. After the heat-shock, embryos were allowed to develop at 28.5°C.

## BrdU and Pharmacological inhibitors

BrdU incorporation, Hydroxyurea and Aphidicolin treatment (inhibition of proliferation) was performed as described in (*Aman et al., 2011*). Rat anti-BrdU (Accurate; OBT0030G) was used at a dilution of 1:500. To visualize nuclei embryos were placed into 0.1 ng/mL DAPI (Invitrogen). All chemical inhibitors were diluted in DMSO (final concentration of 1%) to concentrations as indicated in the text. Used drugs were purchased from Tocris, USA: Fgfr1a inhibitors PD173074 and SU5402, MAPK/ERK inhibitor PD0325901, GSK-3$\beta$ inhibitor BIO and $\gamma$-secretase inhibitor DAPT.

## Time-lapse imaging

Time-lapse imaging was performed as described (*Lush and Piotrowski, 2014*).

## Image analysis

### Cell shape/volume analysis

Individual cells were reconstructed in 3D using Bitplane's Imaris software. Cells were reconstructed manually by drawing the cell boundary in every single z-slice using Imaris' surface function. After the cell boundary was drawn in every z-slice, a surface was created, and the cell volume was exported for analysis.

For illustrative purposes, the cell surfaces were exported, oriented with the cell constriction upward, and placed side by side for comparison. Every attempt was made to export a view of the cells at the same scale, and at an angle, which shows the most representative perspective.

### Apical constriction analysis (area)

Rosette constrictions for each cell were analyzed using the ratio of two area measurements in 2D. First we measured the area of the bounding box of the smallest discernable constriction. Second we measured the area of the bounding box of the largest part of the cell. We then used the ratio of those two measurements to determine how much the cell is constricting on the apical side with respect to the bulk of its structure.

### Apical constriction analysis (volume)

Rosette constrictions were analyzed using the Imaris surface function. Because so much membrane from multiple cells comes together at a single point, constrictions are the brightest feature in the image. A global threshold was set, selecting the brightest portion of the image. Smaller insignificant features were eliminated using a size threshold, leaving only the largest and brightest rosette constrictions.

## Cell counts

Cell counting was performed in Fiji software where every DAPI positive nuclei was counted by hand using a Wand tool. An average number of cells between different samples was calculated and plotted with the standard error bars. Additionally, Student's *t* tests were performed in Microsoft Excel software to assess the significance between the samples.

## Sequencing data analysis

For the RNASeq experiment we cut tails from 36 hpf *Tg(cldnb:lynGFP)* embryos and isolated primordium and neuromast cells by FACS. Reads were mapped to the *Danio rerio* genome version danRer10 from the University of California, Santa Cruz (UCSC) using TopHat version 2.0.13, with gene annotations from Ensembl 84. RNASeq analysis and Gene Ontology (GO) Enrichment analysis was performed in the R environment using the bioconductor packages edgeR, topGO, and biomaRt.

Reads counted on Ensembl transcripts from UCSC using HTSeq version 0.6.1 were analyzed with edgeR to generate P values to form comparisons between wildtype and NICD embryos. Of the 32,105 genes detected, only 19,643 genes with a sum across all samples of at least three reads per million were considered for the analysis. P values were adjusted for multiple hypothesis testing using the Benjamini–Hochberg method. The fragments per kilobase of transcript per million mapped reads (FPKM) values were generated using Cufflinks version 2.2.1. To define differentially expressed genes in the NICD embryos, genes were selected with the following criteria: log2 ratio >0.5 with p value <0.01 and log2 ratio <−0.5 with p value <0.01, resulting in 187 genes up- and 502 genes downregulated, respectively. The necessary GO IDs were obtained from Ensembl using biomaRt. Of the 19,643 expressed genes, 11,757 were associated with GO terms. Therefore, only 117 of the 187 genes and 323 of the 502 genes were used in the GO analysis.

## Data access

The whole-genome sequence data from this study have been submitted to the NCBI Gene Expression Omnibus (GEO; http://www.ncbi.nlm.nih.gov/geo/), accession number GSE86571.

## Acknowledgements

We would like to thank the Piotrowski lab members and Drs. Krumlauf, Gibson, Li and Trainor for stimulating discussions and Dr. Mark Lush and Joaquin Navajas Acedo for critically reading the manuscript. We are thankful to Boris Rubinstein for a help with mathematical analysis. We are grateful to Dr. Ajay Chitnis for *mib1*[*ta52b*] mutant, Dr. Bruce Riley for *Tg(hsp70:atoh1a)*[*x20*], Dr. Virginie Lecaudey for the *shroom3* morpholino and in situ probe, Dr. Brian Link for the UASDNshroom3 plasmid, Dr. Ashley Bruce for the *claudinE* in situ probe, Dr. Anand Chandrasekhar for the *cdh2*[*hi3644*] fish line and Xin Gao for help with the RNASeq analysis. We would also like to thank Helena Boldt for excellent technical support. We are also grateful to Mark Miller for help with illustrations and the Stowers Aquatics facility for excellent help with fish husbandry. We are particularly thankful to the Stowers Institute for Medical Research for Funding.

## Additional information

### Funding

| Funder | Grant reference number | Author |
| --- | --- | --- |
| Stowers Institute for Medical Research | Institutional support | Tatjana Piotrowski |

The funders had no role in study design, data collection and interpretation, or the decision to submit the work for publication.

### Author contributions

AK-G, Conceptualization, Data curation, Formal analysis, Validation, Investigation, Visualization, Methodology, Writing—review and editing; RY, AA, Investigation; RA, Software, Investigation, Visualization; RJ, Resources, Data curation, Formal analysis; DN, HK, Resources; MM, Resources, Investigation; TP, Conceptualization, Supervision, Funding acquisition, Investigation, Writing—original draft, Project administration

### Author ORCIDs

Tatjana Piotrowski, ⓘ http://orcid.org/0000-0001-8098-2574

### Ethics

Animal experimentation: All experiments were performed according to guidelines established by the Stowers Institute IACUC review board (IACUC protocol 2014-0129)

## Additional files

### Major datasets

The following dataset was generated:

| Author(s) | Year | Dataset title | Dataset URL | Database, license, and accessibility information |
|---|---|---|---|---|
| Kozlovskaja-Gumbrienė A, Yi R, Alexander R, Aman A, Jiskra R, Nagelberg D, Knaut H, McClain M, Piotrowski T | 2016 | Proliferation-independent regulation of organ size by Fgf/Notch signaling | https://www.ncbi.nlm.nih.gov/geo/query/acc.cgi?acc=GSE86571 | Publicly available at the NCBI Gene Expression Omnibus (accession no: GSE86571) |

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
