## [Decision Letter]

Thank you for submitting your article "Proliferation-independent regulation of organ size by Notch signaling" for consideration by *eLife*. Your article has been reviewed by three peer reviewers, and the evaluation has been overseen by Tanya Whitfield as the Reviewing Editor and Marianne Bronner as the Senior Editor. The following individual involved in review of your submission have agreed to reveal their identity: Ajay B. Chitnis (Reviewer #2).

The reviewers have discussed the reviews with one another and the Reviewing Editor has drafted this decision to help you prepare a revised submission.

Summary:

This manuscript provides a detailed assessment of the role of Notch signaling in establishing neuromast size in the zebrafish lateral line. Previous studies from the Chitnis lab (Matsuda et al. 2010) described how loss of Notch signaling in mib1 mutants triggers problems in the morphogenesis of the lateral line primordium, including instability of deposited neuromasts and eventual disintegration of the primordium. In contrast to what was shown in this previous study, Kozlovskaja-Gumbriene et al. from the Piotrowski Lab have now shown, using a transgenic line expressing an active form of Notch (NICD), that broad activation of Notch in the primordium formation results in larger and fewer proneuromasts in the migrating primordium. This key observation now reveals a previously unappreciated, more direct role for Notch signaling in determining morphogenesis of epithelial rosettes. While this study confirms many observations made previously by Matsuda et al., the direct comparison of both gain and loss of function effects of Notch signaling allows them to go further by demonstrating the role of Notch signaling in inhibiting Wnt signaling and in acting downstream of FGF signaling to determine morphogenesis of epithelial rosettes.

The study also demonstrates that Notch has a direct role in determining formation and stability of epithelial rosettes, and now provides a clearer explanation for the early phenotype in Mib1 mutants associated with deposition of smaller unstable neuromasts. The study also reveals an important additional role played by Notch ligands expressed in the central Atoh1a expressing cell in both nascent and maturing proneuromasts. In addition to preventing neighboring cells from expressing *atoh1a*, activation of Notch contributes to formation of stable apical constrictions in neighboring cells. Contrary to previous studies that suggested FGF signaling-dependent *shroom3* expression is required for morphogenesis of epithelial rosettes, this study shows that Shroom's role is dispensable.

For the most part, the reviewers found that the authors' conclusions are well supported by the data, and that the study provides an important mechanistic advance for the field. However, the reviewers had a number of comments where additional detail or discussion could help to clarify the findings. Most of these suggestions require relatively minor changes to the text or schematic diagrams. In a few places, additional quantitative analysis would help to support the conclusions. In addition, there were some places where it was felt that the conclusions were over-stated, and should be toned down.

Essential revisions:

1) Ensure descriptions of cell shape are accurate, and distinguish between quantitative and qualitative data (rev. 1).

2) State expected outcomes for the hypotheses that are being tested (rev. 1).

3) Include quantitative data to address the comment about mosaicism from reviewer 1, or tone down the conclusions from this section.

4) Consider changing the title (rev. 2).

5) Add a caveat about the control of expression of Ecad by Atoh1a (rev. 2).

6) Discuss or rebut the suggestion that control of *wnt10a* expression may be indirect (rev. 2).

7) Clarify the schematic of the 1st proneuromast (rev. 2).

8) Address and clarify the relationship between primordium and neuromast size in the NICD overexpression experiments (rev. 3).

9) State exactly how the cell counts were done, and provide additional data to clarify the relationship between NICD, Yap1 and primordium cell number, if possible (rev. 3).

10) Ensure that the schematics are close representations of the data shown (rev. 3).

11) Make the manuscript shorter and more concise where possible (rev. 3).

Detailed versions of the reviews are appended below for further information.

*Reviewer #1:*

1) In the subsection “A reduction in Notch leads to loss of cell-cell adhesion and apical constrictions”, Figure 6: The data presented in Figure 6 are difficult to interpret. First, the rows of isolated cells in panels A-C are difficult to relate to their counterparts in the intact images. Cell shapes in L1 appear similar in NICD and mib embryos but are quite distinct from wt, yet are described as normal. Cell constrictions in L2 are said to be lost in mib mutants. This is evident in some cells (e.g. blue and pink) but less so in others. And the pink cell in L1 in NICD also appears to lack an obvious apical constriction. Somehow interpretations in this part of the paper seem highly subjective. Note, Figure 6 is labeled to direct attention to Video 5, whereas the text mentions Video 4. Figure 6 marks "constriction volume", but it is not clear what this means – and is this quantitative as the term suggests, or strictly qualitative as the images suggest? In the last sentence it is stated that the phenotypic defects in mib mutants become progressively more severe "because of maternal rescue". It would make more sense to say the phenotype worsens as maternal stores become depleted.

2) In the subsection “Notch-positive cells self-organize into rosettes”, Figure 7: Mosaic analysis is often a powerful means of distinguishing between competing hypotheses, but it is important to clearly articulate expected outcomes that might support one hypothesis over the other. In this case expectations are not discussed, nor is it clear whether data support the "signaling center" hypothesis or the "altered cell adhesion" hypothesis. With either model, one might expect that the number of wt cells recruited into a neuromast might vary depending on the number of transgenic cells present. Yet there was no attempt to correlate neuromast size with the percentage of wt vs. transgenic cells. Moreover, it is concluded that wt cells were "never recruited to make neuromasts larger than they would normally be". The basis for this is unclear. The image in Figure 7" appears to show a larger than normal neuromast that includes a mix of both wt and NICD cells, in apparent contradiction of the conclusion. Size measurements and cell counts (wt & NICD) are needed to make a convincing argument. Other data in Figure 7 are more convincing. On a technical note, the high mag image in 7G' is inverted horizontally relative to the low mag image in 7G.

*Reviewer #2:*

While overall the paper is excellent and reveals another example of a context in which Notch has a more general role in promoting an epithelial morphology, there are a few details related to interpretation or emphasis with which I did not agree. These issues are enumerated below.

1) I thought the title of the paper, which emphasizes a role for Notch signaling in organ size formation is misleading. While in the gain of function experiments many more cells can be recruited to become part of an epithelial rosette, as noted above, this gain of function phenotype does not represent that pattern of Notch activation or epithelial rosette formation seen under normal circumstances, where formation of rosettes is beautifully coordinated by a central cell that becomes the source of FGFs and Notch ligands. Instead, I believe the paper provides excellent evidence for a role for Notch signaling in not only preventing neighboring cells from expressing *atoh1a* but also in determining formation of stable epithelial rosettes by activating Notch in the neighbors. The size of these rosettes may, under physiological conditions, still be determined by FGF signaling-though, now as we have learnt from this study, not only by *shroom* or MAPK dependent mechanisms as previously described but in addition by Notch signaling. As the authors themselves state, by determining Notch3 expression, FGF signaling may still play a critical role in determining which how many cells are recruited to become part of the rosette by Notch signaling. The gain of function experiments with NICD do, however, show how in the absence of all patterning determined by interactions between Wnt and FGF signaling systems, Notch signaling can independently determine a particular configuration of epithelial rosettes in the primordium.

2) In the subsection “Notch signaling upregulates cell-cell adhesion molecules and tight junction genes”, second paragraph: the authors show that e-cadherin expression is promoted by Notch signaling. While data in this paper supports this conclusion, it remains possible that in addition to being promoted by Notch signaling, its expression is inhibited by Atoh1a as suggested by Matsuda et al., since knock-down of *atoh1a* leads to some recovery of e-cadherin expression in *mib1* mutants.

3) "To test if Notch signaling restricts wnt10a expression we treated *apc* mutant embryos with the γ-secretase inhibitor DAPT (Figure 9). Indeed, wnt10a is now expressed in many more trailing cells demonstrating that Notch inhibits wnt10a, a feedback interaction not previously described in the primordium (Figure 8). This negative interaction also explains why *wnt10a* is restricted to a single central cell in the first forming proneuromast of a wildtype primordium (Figure 9)".

While this observation superficially supports the assertion of the authors that Notch signaling restricts Wnt10a expression, I feel what is happening here might be more complicated and only indirectly related to Notch signaling. One problem is that the broad spread of *wnt10a* expression seen in *apc* mutants is only seen following 20 hours of exposure to DAPT. Does it spread with a shorter treatment? If not, I suspect that the expansion of *wnt10a* expression is the indirect effect of something that takes a while to develop. One possibility is that it is related to progressive slow expansion in the number of *atoh1a* expressing cells. As Atoh1a inhibits expression of the FGF receptor, it would therefore permit Wnt signaling in this cell, which is so close to the source of Wnt signals. This may explain why transiently or in an oscillatory pattern Wnt10a is expressed in a spot just adjacent to the leading cloud of *wnt10* expression, where it is expressed under normal circumstances. Before the authors can conclude that the restricted pattern is determined by lateral inhibition they should determine if knock-down of *atoh1a* prevents the expansion. Alternatively, they can skip the experiment and suggest this alternate hypothesis.

4) The schematic for signaling in the 1st proneuromast (most recently formed?) has a number of problems. Notch3 is not being activated by DeltaA/D but by Ras-Mapk which confuses transcriptional regulation of Notch3 and actual activation by its ligand. If there are details the author feels are not well known at this stage they should consider dropping the schematic of early proneuromast or revise it to make it accurate. I feel what’s most clear from their paper is what is presented for the maturing proneuromast.

*Reviewer #3:*

This paper studies the role of increasing Notch signalling on lateral line development. How organ size is regulated by balancing cell proliferation and cell differentiation is an important question and the lateral line has been a good experimental model to address these questions. The authors deepen into them, and here they propose that in contrast to Wnt signaling that increases neuromast cell numbers by increasing cell proliferation, Notch regulates organ size independently of cell proliferation and by regulating cell adhesion and rosette number in the primordium. The authors combine several approaches and provide an extensive analysis of Notch effects. The work provides interesting, but not fully novel data and some of the conclusions are precipitated and not supported by their data. The link between the effects on cell adhesion and NMs size is not clear. On a side, there are several functional experiments addressing NICD effects on NMs size. On another set of experiments, the authors address the effects of NICD on rosette formation analyzing the primordium without showing the effect on deposited NMs size. In overall, the paper is very long, difficult to follow and without a clear-cut idea of the Notch role on NMs size.

1) Role of NICD in proliferation

The authors make strong emphasis in demonstrating that Notch overactivation increases neuromast size independent on effects on cell proliferation. The authors claim that the size of NICD NM is independent of primordium size (figure legend, Figure 1—figure supplement 1) and this is not fully supported by their data. When overexpressing NICD early by combining cdnB:gal4 with UAS:NICD, the size of the primordium is bigger (while ganglion smaller) and in this condition, the L1 instead of containing approx 35 cells contains 110. Through development, as pLLP decreases its size, the NM size also decreases, indicating that the size of the primordium has an influence on NM size. If NICD is overexpressed after 25 hpf, the L1 or L2 NM only increases its size from 35 cells to 45-55. The authors suggest that as the size of the primordium in this condition is not bigger (due to the effect of ganglion-primordium fate switch) but the NM size is still bigger, the size of the primordium is not the cause of NM increase. However, in this later condition, the effect over NM size is much milder, indicating that if NICD is overactivated early, this has a very strong effect of first neuromasts deposited. How the authors explain that, if the size of the primordium has no effect on NM size, the early effects are much stronger than in the hs:NICD. It is not clear why all the data is shown in embryos in which NICD has been manipulated early instead of hsp:NICD to overcome possible effects of initial primordium size. The authors should also provide data on the size of the primordium in hs:NICD experiments.

Again, the authors state "whereas the increase in the size of NICD neuromasts is controlled by a proliferation-independent process within the primordium".

The difference in the proliferation status in NICD NMs and *apc* NMs is clear but the authors also show that the BrdU index at the NICD primordium is higher than controls and similar to *apc* mutants (Figure 3, although stated that does not change). Could the increased proliferation at the primordium affect deposited NMs size?

The authors also show that the neuromast size in NICD is independent of *yap1*, while there is a strong reduction in primordium cell numbers in NICD-yap MO embryos (Figure 2), the size of the NMs is not significantly reduced in this condition. In this particular experiment the deviation bars are higher, thus this mild reduction might become significant if increasing the number of embryos analysed. This is important because is one of the major conclusions of the paper. I suggest to repeat this experiment and add more embryos. Throughout the paper is not clear, which NMs are counted for cell numbers and at which stage the counting is done.

Figure 1 shows that the relative difference of size of wt and NICD NM is reduced overtime, however in Figure 3 this difference is kept. Please explain incongruence. As part of this, it is stated in the text that NICD neuromasts do not significantly grow in cell number (Figure 3) but instead the NICD NM size increases over time as shown in this figure. How this happens if cell proliferation decreases so drastically in NM (Figure 3)?

In the schematic representation of the primordium shown in Figure 4, the NICD primordium is not bigger. However, Figure 6 (primordia shown are at different stages!) shows a much bigger primordium in NICD. In particular, the size of the leading part is bigger (not depicted in the drawing). Again, the changes in size of the primordium and in particular the ratio of leading and trailing cells might be relevant for the neuromast size of NICD embryos. This is very little discussed or addressed.

2) Role of NICD in cell fate specification

This section is confusing. If by cell fate the authors refer to the switch between the 4 cell types defined in the primordium, it is clear that there is not a major cell fate switch. The authors conclude that NICD causes the allocation of support cells into fewer but larger proneuromasts and this is clear. But, as the authors mention later, within the primordium, the first forming proneuromast and more trailing proneuromasts are in different stages of cell fate specification due to different activities of FGF signaling. Since Notch affects FGF and Wnt signaling, it is possible that the proneuromast lost is converted to the fate of the other proneuromasts. It is possible that the disappearance of a proneuromast is due to the cell signaling effects and changes of proneuromast commitment stage and not due to increased recruitment of Rock2a that would be secondary. Please discuss this.

3) Cell adhesion:

The authors claim that the main cause of increased NICD neuromast size is their increased adhesivity, more cells sticking together, but not effects on rosette formation are observed when blocking some cell adhesion molecules. In addition, the conclusion is also taken that interneuromasts cells cluster more together in NICD, but these are cells not related to the focus of the study. The conclusion is strong for the little evidences on the cell adhesion and I would suggest to tune-down the conclusions. As in NICD more cells compose a proneuromast, it is obvious that the E-cadherin staining is going to be larger (Figure 7). Moreover, in the NICD primordium one of the rosette disappears making the proneuromast larger, but if cell adhesivity is increased, why not more rosettes are formed instead? The authors suggest that increased cell adhesion, affects the morphology of the organ to achieve minimal surface tension. If data from microsurgery experiments to measure of surface tensions in wt and NICD cells are available, this would be an interesting addition, but is not essential.

4) Notch and FGF/wnt signaling

The epistatic data showing that Notch is downstream of Wnt and FGF signaling is clear. In parallel Notch negatively feeds-back in FGF signalling by reducing *fgf10a, fgf3* expression (not clear about *pea3*). On the other hand, NICD reduces the levels of Lef1, *sef1, dkk1b* and *wnt10a* at the leading region. The effects on Wnt signalling are quite strong and previous reports have analysed the influence of the disruption of Wnt in proliferation and neuromast deposition. The authors make strong emphasis showing that the effects on NMs size is different that the phenotype caused by *apc* mutants. However, since NICD modifies the balance of FGF/Wnt signalling, it is not clear that Notch and Wnt are not linked in the regulation of primordium size and differentiation. In NICD embryos, what happens if Wnt signaling is raised by crossing with *apc* mutants? If this is known, please discuss.

---

## [Author Response]

*[…] Reviewer #1:*

*1) In the subsection “A reduction in Notch leads to loss of cell-cell adhesion and apical constrictions”, Figure 6: The data presented in Figure 6 are difficult to interpret. First, the rows of isolated cells in panels A-C are difficult to relate to their counterparts in the intact images.*

We now assigned individual colors for every cell to make it easier to find their positions in the intact primordium. Also, we added an additional panel with colored cells in the proneuromasts and without the cldnB:GFP signal in the background to again make it easier to locate each cell (Figure 6).

*Cell shapes in L1 appear similar in NICD and mib embryos but are quite distinct from wt, yet are described as normal. Cell constrictions in L2 are said to be lost in mib mutants. This is evident in some cells (e.g. blue and pink) but less so in others. And the pink cell in L1 in NICD also appears to lack an obvious apical constriction. Somehow interpretations in this part of the paper seem highly subjective.*

We agree with the reviewer and have added quantifications of the apical constriction ratios (new Figure 6). These data show that cells in the last proneuromast in the mib primordium constrict less than WT or NICD cells. We defined constriction as a ratio between the area of the portion of the cell that has the smallest discernable constriction, and the area in the slice that has the largest area. While the individual cells within one graph are scaled equally with respect to each other, no scaling comparison can be made between different samples. This is due to the limitations of the Imaris scale bar function, rooted in the complexity of displaying 3D data.

*Note, Figure 6 is labeled to direct attention to Video 5, whereas the text mentions Video 4.*

We now have changed the labels to indicate the correct Video 4.

*Figure 6 marks "constriction volume", but it is not clear what this means – and is this quantitative as the term suggests, or strictly qualitative as the images suggest?*

We have now included the quantification for the constriction volume analysis (new Figure 6). The bar graph shows that the subtraction value of the constriction volumes of R3 (more mature proneuromasts) and R2 (less mature proneuromasts) is positive for WT primordia, whereas it is negative for mib primordia. Therefore, the constriction volumes decrease in mib as proneuromasts mature and before they are deposited.

*In the last sentence it is stated that the phenotypic defects in mib mutants become progressively more severe "because of maternal rescue". It would make more sense to say the phenotype worsens as maternal stores become depleted.*

We have rephrased the last sentence of this paragraph accordingly.

*2) In the subsection “Notch-positive cells self-organize into rosettes”, Figure 7: Mosaic analysis is often a powerful means of distinguishing between competing hypotheses, but it is important to clearly articulate expected outcomes that might support one hypothesis over the other. In this case expectations are not discussed, nor is it clear whether data support the "signaling center" hypothesis or the "altered cell adhesion" hypothesis. With either model, one might expect that the number of wt cells recruited into a neuromast might vary depending on the number of transgenic cells present. Yet there was no attempt to correlate neuromast size with the percentage of wt vs. transgenic cells.*

Our new quantification strongly favors the “altered cell adhesion” hypothesis over the “signaling center” hypothesis. Specifically, if NICD cells acted as a signaling center we would find enlarged mosaic neuromasts with just one or few NICD cells in it and with more than a normal number of wildtype cells recruited. However, our new quantification shows that larger mosaic neuromasts are observed only when NICD cells contribute to the mosaic neuromast. Moreover, the number of wildtype host cells remains similar between two transplantation conditions (Figure 7’). Together these findings strongly support the hypothesis that larger neuromasts are formed due to altered cell adhesion between NICD cells and no wildtype cells are recruited.

*Moreover, it is concluded that wt cells were "never recruited to make neuromasts larger than they would normally be". The basis for this is unclear. The image in Figure 7" appears to show a larger than normal neuromast that includes a mix of both wt and NICD cells, in apparent contradiction of the conclusion. Size measurements and cell counts (wt & NICD) are needed to make a convincing argument. Other data in Figure 7 are more convincing.*

See above. The new data verifies that wildtype neuromast size does not exceed the normal number of cells when wildtype cells are transplanted into it (cell number in Figure 7’ is similar to wildtype neuromast size in Figure 1, which is around 38 cells). Indeed, an increase in the neuromast size from on average 38 cells to 45 cells is due to the number of NICD cells present in it (Figure 7’).

*On a technical note, the high mag image in 7G' is inverted horizontally relative to the low mag image in 7G.*

We would like to thank the reviewer for pointing this out. We now have corrected the inversion in Figure 7’.

*Reviewer #2:*

*While overall the paper is excellent and reveals another example of a context in which Notch has a more general role in promoting an epithelial morphology, there are a few details related to interpretation or emphasis with which I did not agree. These issues are enumerated below.*

*1) I thought the title of the paper, which emphasizes a role for Notch signaling in organ size formation is misleading. While in the gain of function experiments many more cells can be recruited to become part of an epithelial rosette, as noted above, this gain of function phenotype does not represent that pattern of Notch activation or epithelial rosette formation seen under normal circumstances, where formation of rosettes is beautifully coordinated by a central cell that becomes the source of FGFs and Notch ligands. Instead, I believe the paper provides excellent evidence for a role for Notch signaling in not only preventing neighboring cells from expressing atoh1a but also in determining formation of stable epithelial rosettes by activating Notch in the neighbors. The size of these rosettes may, under physiological conditions, still be determined by FGF signaling-though, now as we have learnt from this study, not only by shroom or MAPK dependent mechanisms as previously described but in addition by Notch signaling. As the authors themselves state, by determining Notch3 expression, FGF signaling may still play a critical role in determining which how many cells are recruited to become part of the rosette by Notch signaling. The gain of function experiments with NICD do, however, show how in the absence of all patterning determined by interactions between Wnt and FGF signaling systems, Notch signaling can independently determine a particular configuration of epithelial rosettes in the primordium.*

The reviewer made a convincing argument that in wildtype embryos FGF signaling is likely still crucial for determining rosette size by regulating the number of cells that turn on Notch signaling. We have therefore changed the title to: ‘Proliferation-independent regulation of organ size by FGF/Notch signaling’.

*2) In the subsection “Notch signaling upregulates cell-cell adhesion molecules and tight junction genes”, second paragraph: the authors show that e-cadherin expression is promoted by Notch signaling. While data in this paper supports this conclusion, it remains possible that in addition to being promoted by Notch signaling, its expression is inhibited by Atoh1a as suggested by Matsuda et al., since knock-down of atoh1a leads to some recovery of e-cadherin expression in mib1 mutants.*

To test Atoh1a function in E-cadherin regulation we overexpressed Atoh1a in the lateral line by heat-shock, which resulted in elevated *e-cadherin* expression in the primordium (Figure 7CC’’). Thus, *e-cadherin* is not inhibited by *atoh1a*. In addition, *atoh1a* overexpression leads to increased *her4* expression in the primordium, which suggests that Atoh1a positively regulates Notch signaling in the lateral line which then leads to the induction of *e-cadherin*. We added these findings to the second paragraph of the subsection “Notch signaling upregulates *e-cadherin* expression”.

*3) "To test if Notch signaling restricts wnt10a expression we treated apc mutant embryos with the γ-secretase inhibitor DAPT (Figure 8). Indeed, wnt10a is now expressed in many more trailing cells demonstrating that Notch inhibits wnt10a, a feedback interaction not previously described in the primordium (Figure 9). This negative interaction also explains why wnt10a is restricted to a single central cell in the first forming proneuromast of a wildtype primordium (Figure 8).*

*While this observation superficially supports the assertion of the authors that Notch signaling restricts Wnt10a expression, I feel what is happening here might be more complicated and only indirectly related to Notch signaling. One problem is that the broad spread of wnt10a expression seen in apc mutants is only seen following 20 hours of exposure to DAPT. Does it spread with a shorter treatment? If not, I suspect that the expansion of wnt10a expression is the indirect effect of something that takes a while to develop. One possibility is that it is related to progressive slow expansion in the number of atoh1a expressing cells. As Atoh1a inhibits expression of the FGF receptor, it would therefore permit Wnt signaling in this cell, which is so close to the source of Wnt signals. This may explain why transiently or in an oscillatory pattern Wnt10a is expressed in a spot just adjacent to the leading cloud of wnt10 expression, where it is expressed under normal circumstances. Before the authors can conclude that the restricted pattern is determined by lateral inhibition they should determine if knock-down of atoh1a prevents the expansion. Alternatively, they can skip the experiment and suggest this alternate hypothesis.*

To address the reviewer’s concern that *wnt10a* might be expanding in DAPT-treated primordia due to a gradual FGF loss and consequent expansion of Wnt (Figure 8’), rather than de-repression by the loss of Notch, we tested if Notch is sufficient to restrict *wnt10a* in the presence of FGF inhibitor. Our experiment shows that *wnt10a* expression is restricted to the leading domain in the NICD primordium, even in the presence of FGF inhibitor (Figure 8’), strongly suggesting that Notch inhibits *wnt10a* independently ofFGF/Atoh1a. We have now added a schematic representation of the status of *wnt10a, pea3, atoh1a* and *notch3* in primordia in which Wnt, Notch and FGF are manipulated individually or in combination (Figure 8). *wnt10a* only expands in primordia in which Notch is depleted, irrespective of the activation status of FGF (*pea3*) or *atoh1a*. We therefore conclude that Notch inhibits Wnt signaling in parallel to these factors. Changes were made in the fourth and fifth paragraphs of the subsection “Notch signaling is a component of the Wnt/Fgf signaling network that coordinates lateral line morphogenesis”.

*4) The schematic for signaling in the 1st proneuromast (most recently formed?) has a number of problems. Notch3 is not being activated by DeltaA/D but by Ras-Mapk which confuses transcriptional regulation of Notch3 and actual activation by its ligand. If there are details the author feels are not well known at this stage they should consider dropping the schematic of early proneuromast or revise it to make it accurate. I feel what’s most clear from their paper is what is presented for the maturing proneuromast.*

To address reviewer’s comment, we changed several aspects of Figure 9. First, in the Figure 9 we isolated hair cell progenitor cell in green to better illustrate a valid point that Notch is not transcriptionally activated by deltaD/A but rather by lateral-inhibition. Also, to convey the same message we used a dashed line from deltaD/A instead of a solid line towards Notch in the dark blue cell in Figure 9.*Reviewer #3:*

*This paper studies the role of increasing Notch signalling on lateral line development. How organ size is regulated by balancing cell proliferation and cell differentiation is an important question and the lateral line has been a good experimental model to address these questions. The authors deepen into them, and here they propose that in contrast to Wnt signaling that increases neuromast cell numbers by increasing cell proliferation, Notch regulates organ size independently of cell proliferation and by regulating cell adhesion and rosette number in the primordium. The authors combine several approaches and provide an extensive analysis of Notch effects. The work provides interesting, but not fully novel data and some of the conclusions are precipitated and not supported by their data. The link between the effects on cell adhesion and NMs size is not clear. On a side, there are several functional experiments addressing NICD effects on NMs size. On another set of experiments, the authors address the effects of NICD on rosette formation analyzing the primordium without showing the effect on deposited NMs size. In overall, the paper is very long, difficult to follow and without a clear-cut idea of the Notch role on NMs size.*

We have shortened and rephrased some of the sections to make the paper easier to read and to emphasize that our data shows that Notch signaling cell-autonomously upregulates apical junction complex genes which induce apical constriction and cell adhesion. These changes in cell adhesion and apical constriction lead to a coalescence of support cells into larger neuromasts in the absence of proliferation.

*1) Role of NICD in proliferation*

*The authors make strong emphasis in demonstrating that Notch overactivation increases neuromast size independent on effects on cell proliferation. The authors claim that the size of NICD NM is independent of primordium size (figure legend, Figure 1—figure supplement 1) and this is not fully supported by their data. When overexpressing NICD early by combining cdnB:gal4 with UAS:NICD, the size of the primordium is bigger (while ganglion smaller) and in this condition, the L1 instead of containing approx 35 cells contains 110. Through development, as pLLP decreases its size, the NM size also decreases, indicating that the size of the primordium has an influence on NM size. If NICD is overexpressed after 25 hpf, the L1 or L2 NM only increases its size from 35 cells to 45-55. The authors suggest that as the size of the primordium in this condition is not bigger (due to the effect of ganglion-primordium fate switch) but the NM size is still bigger, the size of the primordium is not the cause of NM increase. However, in this later condition, the effect over NM size is much milder, indicating that if NICD is overactivated early, this has a very strong effect of first neuromasts deposited. How the authors explain that, if the size of the primordium has no effect on NM size, the early effects are much stronger than in the hs:NICD.*

We would like to thank the reviewer for this thoughtful comment. We agree that the primordium size has an influence on how big neuromasts can get and we are discussing this point now in the text (subsection “Constitutive activation of Notch or Wnt signaling generates larger sensory organs”, last paragraph). However, we show that the primordium size only limits the maximum size that NICD neuromasts can reach but it does not cause the larger neuromast size in NICD embryos. In support of this conclusion, *amotl2a* mutant embryos possess much larger primordia but the neuromasts do not increase in size (Agarwala et al., 2015). However, the primordium size limits the maximum size of the NICD neuromasts, cldnB:gal4xUAS:NICD neuromasts get smaller as the primordium gets smaller. Importantly, even though NICD primordia eventually reach the same size as wildtype primordia (Figure 1 [32 hpf time point]), they keep on generating larger neuromasts (Figure 1).

*It is not clear why all the data is shown in embryos in which NICD has been manipulated early instead of hsp:NICD to overcome possible effects of initial primordium size. The authors should also provide data on the size of the primordium in hs:NICD experiments.*

The hs:NICD line only became available to us after the majority of experiments had already been performed. As the experiments with the hs:NICD line supported our findings of experiments that we performed with the UAS:NICD line, we did not redo all experiments in the hs:NICD line.

Again, the authors state "whereas the increase in the size of NICD neuromasts is controlled by a proliferation-independent process within the primordium".

*The difference in the proliferation status in NICD NMs and apc NMs is clear but the authors also show that the BrdU index at the NICD primordium is higher than controls and similar to apc mutants (Figure 3, although stated that does not change). Could the increased proliferation at the primordium affect deposited NMs size?*

We agree that there is a slight increase in proliferation in NICD primordia in Figure 3. However, this change is not statistically significant and has no effect on the primordium cell number in Figure 3. The results of the experiments in which we reduced proliferation using HUA and Aph (Figure 3) demonstrate that, while the inhibition of proliferation reduces the number of primordium cells significantly in NICD and *apc* primordia (Figure 3) it does not significantly rescue the NICD neuromast size, which is still significantly higher than wildtype neuromast size (Figure 3). *apc* neuromasts, on the other hand, are reduced back to wildtype size in the absence of proliferation. Together these data show that proliferation does not cause the significant increase in neuromast size in NICD neuromasts.

*The authors also show that the neuromast size in NICD is independent of yap1, while there is a strong reduction in primordium cell numbers in NICD-yap MO embryos (Figure 2), the size of the NMs is not significantly reduced in this condition. In this particular experiment the deviation bars are higher, thus this mild reduction might become significant if increasing the number of embryos analysed. This is important because is one of the major conclusions of the paper. I suggest to repeat this experiment and add more embryos.*

As it was suggested by the reviewer we repeated NICD+yap1MO experiment and plotted results from a new experiment in the Figure 2 (old results were not added to this quantification). Indeed, by analyzing more embryos we see a significant reduction in NICD+yap1MO L1 and about to be deposited L2 sizes as compared to NICD control. Nevertheless, NICD+yap1MO neuromasts are still significantly larger than WT control neuromasts or WT embryos injected with yap1MO (Figure 2). Therefore, even though the NICD+yap1MO primordium size is rescued back to a wildtype size (Figure 2), NICD neuromasts remain too large and this increase in size is *yap1*-independent. We believe that the reduction in neuromast size in *yap1* morpholino injected embryos is caused by the effect of *yap1* loss in the primordium. Changes to the text were made in the last paragraph of the subsection “The increased organ size after Notch and Wnt activation is independent of *yap1*”.

*Throughout the paper is not clear, which NMs are counted for cell numbers and at which stage the counting is done.*

We apologize for the oversight and have now added this information to the following figures:

Figure 2: L2 proneuromast (still inside the primordium) at 30hpf

Figure 2: L1 neuromast at 30hpf

Figure 3: L1 neuromast (different time-points are indicated in the figure)

Figure 3: L3 neuromast at 35hpf

Figure 4: L1 neuromast (different time-points are indicated in the figure)

Figure 5—figure supplement 1: L3 neuromast at 3dpf

Figure 7: L1 neuromast (different time-points are indicated in the figure).

*Figure 1 shows that the relative difference of size of wt and NICD NM is reduced overtime, however in Figure 3 this difference is kept. Please explain incongruence.*

Figure 1 shows the sizes of all posterior lateral line neuromasts (L1-5) along the trunk of the embryo at a single time point (2.5dpf), whereas, Figure 3 describes the growth of the L1 neuromast between 32-56hpf. We have clarified the text, subsection “Constitutive activation of Notch or Wnt signaling generates larger sensory organs”, last paragraph and figure legend.

*As part of this, it is stated in the text that NICD neuromasts do not significantly grow in cell number (Figure 3) but instead the NICD NM size increases over time as shown in this figure. How this happens if cell proliferation decreases so drastically in NM (Figure 3)?*

We agree with the reviewer that NICD L1 neuromast showed on average more cells at 51hpf compared to 26hpf in the old graph, but, importantly, the increase was not statistically significant. We repeated the experiment evaluating L1 neuromast growth over-time in WT, NICD and apc embryos. This time we added more embryos to the quantification. Consistent with our previous result NICD L1 neuromasts do not grow significantly between 32hpf and 56hpf. Therefore, this result is consistent with a significant proliferation decrease in NICD L1 at 35hpf in Figure 3.

*In the schematic representation of the primordium shown in Figure 4, the NICD primordium is not bigger. However, Figure 6 (primordia shown are at different stages!) shows a much bigger primordium in NICD. In particular, the size of the leading part is bigger (not depicted in the drawing). Again, the changes in size of the primordium and in particular the ratio of leading and trailing cells might be relevant for the neuromast size of NICD embryos. This is very little discussed or addressed.*

The reviewer is correct. At 27 hpf (Figure 6) NICD primordia are still significantly larger than wildtype primordia (also see Figure 1). We now generated new data shown in Figure 4 where we describe the NICD primordium cell composition at 26hpf (Figure 4). The mesenchymal domains are similar between WT and NICD primordia at 26hpf, which indeed suggests that the ratio between the amount of leading and trailing cells is changing in the primordium during migration. However, we do not believe that the size of the leading region significantly affects neuromast size, as 26 hpf primordia have a normal cell number in the leading region but deposit larger neuromasts (subsection “The increase in organ size is independent of the role of Notch in cell type specification”, last paragraph).

*2) Role of NICD in cell fate specification*

*This section is confusing. If by cell fate the authors refer to the switch between the 4 cell types defined in the primordium, it is clear that there is not a major cell fate switch.*

We added a sentence to clarify this point: “To test if Notch changes the fate of one cell population into another we counted the number of cells in these four populations in wildtype and NICD primordia.”

*The authors conclude that NICD causes the allocation of support cells into fewer but larger proneuromasts and this is clear. But, as the authors mention later, within the primordium, the first forming proneuromast and more trailing proneuromasts are in different stages of cell fate specification due to different activities of FGF signaling. Since Notch affects FGF and Wnt signaling, it is possible that the proneuromast lost is converted to the fate of the other proneuromasts. It is possible that the disappearance of a proneuromast is due to the cell signaling effects and changes of proneuromast commitment stage and not due to increased recruitment of Rock2a that would be secondary. Please discuss this.*

As mentioned by the reviewer, Notch does not only balance cell fates, but it also inhibits Wnt and thus FGF signaling, which are critically important for primordium patterning. Therefore, the idea that the disruption of primordium patterning by Notch could be transforming the fates of individual proneuromasts into more or less proneuromasts that then aberrantly coalesce is attractive. However, our data shows that irrespective of the activation status of Wnt and FGF in NICD primordia, always larger neuromasts form in the presence of Notch. Specifically, neuromasts are equally well formed when FGF is completely depleted in NICD primordia (FGF inhibitor treatments in Figure 5’) or upregulated (FGF is elevated upon Wnt overexpression in Figure 3). Similarly, the suppression or upregulation of Wnt in NICD embryos does not rescue NICD neuromast size (Figure 9’ and S, S’). In conclusion, in the presence of Notch signaling, the activation status of Wnt or FGF does not influence NICD neuromast size, as Notch acts downstream of these pathways.

*3) Cell adhesion:*

*The authors claim that the main cause of increased NICD neuromast size is their increased adhesivity, more cells sticking together, but not effects on rosette formation are observed when blocking some cell adhesion molecules. In addition, the conclusion is also taken that interneuromasts cells cluster more together in NICD, but these are cells not related to the focus of the study. The conclusion is strong for the little evidences on the cell adhesion and I would suggest to tune-down the conclusions. As in NICD more cells compose a proneuromast, it is obvious that the E-cadherin staining is going to be larger (Figure 7).*

The reviewer makes a valid point, however the fact that the *e-cadherin* domain in the apical wall of single cells is increased (Figure 7) and adherens junctions are significantly longer in TEM images (Figure 7), argues that E-cadherin is upregulated by NICD at the single cell level. Also, our transplantation experiments demonstrate that only transplanted NICD cells cause enlarged neuromasts, arguing that they adhere to each other (Figure 7). The reviewer is correct that lateral line interneuromast cells are a different cell population. However, cell adhesion properties are often tested in different cell types of the same genotype (see hanging drop adhesion assays or transplantation of wildtype and mutant cells into gastrulating zebrafish embryos to test their migratory and clonal behaviors). We therefore believe that interneuromast cell behavior is a good proxy for how neuromast cells behave.

*Moreover, in the NICD primordium one of the rosette disappears making the proneuromast larger, but if cell adhesivity is increased, why not more rosettes are formed instead? The authors suggest that increased cell adhesion, affects the morphology of the organ to achieve minimal surface tension. If data from microsurgery experiments to measure of surface tensions in wt and NICD cells are available, this would be an interesting addition, but is not essential.*

We agree but such experiments are unfortunately beyond the scope of this manuscript.

*4) Notch and FGF/wnt signaling*

*The epistatic data showing that Notch is downstream of Wnt and FGF signaling is clear. In parallel Notch negatively feeds-back in FGF signalling by reducing fgf10a, fgf3 expression (not clear about pea3). On the other hand, NICD reduces the levels of Lef1, sef1, dkk1b and wnt10a at the leading region. The effects on Wnt signalling are quite strong and previous reports have analysed the influence of the disruption of Wnt in proliferation and neuromast deposition. The authors make strong emphasis showing that the effects on NMs size is different that the phenotype caused by apc mutants. However, since NICD modifies the balance of FGF/Wnt signalling, it is not clear that Notch and Wnt are not linked in the regulation of primordium size and differentiation. In NICD embryos, what happens if Wnt signaling is raised by crossing with apc mutants? If this is known, please discuss.*

This point is related to the question raised in 2) with regard to how the effect of Notch on the other signaling pathways might contribute to the NICD phenotype. However, in NICD Wnt signaling is reduced and the neuromasts get bigger, whereas in apc mutants Wnt signaling is increased and the neuromasts also get bigger. This finding suggests that the effect of Notch on neuromast size is not caused by its inhibitory effect on Wnt signaling. We also added the experiment suggested by the reviewer where we activated Wnt signaling in NICD and WT embryos by soaking the embryos in BIO. In BIO-treated NICD embryos neuromasts become even larger than in untreated NICD embryos (Figure 3). This shows that Wnt and Notch have an additive effect and therefore affect neuromast size by different mechanisms, namely increasing proliferation and adhesion (subsection “Activated Notch leads to proliferation-independent neuromast growth”, last paragraph.

The reviewer is correct that one would expect to affect proliferation in NICD primordia if Wnt signaling is reduced. However, we have previously shown that Wnt and FGF together (Aman et al., 2011) regulate proliferation. Possibly, proliferation is still triggered in NICD primordia, as in NICD primordia both Wnt and FGF are reduced